# Bactericidal antibiotic treatment induces damaging inflammation via TLR9 sensing of bacterial DNA

Julia L. Gross [1], Rahul Basu[2], Clinton J. Bradfield [2], Jing Sun[2], Sinu P. John[2], Sanchita Das[3], John P. Dekker [3,4], David S. Weiss [5,6,7] & Iain D. C. Fraser [2,7]

The immunologic consequences of using bactericidal versus bacteriostatic antibiotic treatments are unclear. We observed a bacteriostatic (growth halting) treatment was more protective than a bactericidal (bacteria killing) treatment in a murine peritonitis model. To understand this unexpected difference, we compared macrophage responses to bactericidal treated bacteria or bacteriostatic treated bacteria. We found that Gram-negative bacteria treated with bactericidal drugs induced more proinflammatory cytokines than those treated with bacteriostatic agents. Bacterial DNA – released only by bactericidal treatments – exacerbated inflammatory signaling through TLR9. Without TLR9 signaling, the in vivo efficacy of bactericidal drug treatment was rescued. This demonstrates that antibiotics can act in important ways distinct from bacterial inhibition: like causing treatment failure by releasing DNA that induces excessive inflammation. These data establish a novel link between how an antibiotic affects bacterial physiology and subsequent immune system engagement, which may be relevant for optimizing treatments to simultaneously clear bacteria and modulate inflammation.

Antibiotics are a fundamental cornerstone of modern medicine. In addition to their use in the treatment of potentially serious infections in otherwise healthy individuals, they have enabled the development of modern approaches in surgery, chemotherapy, and transplantation, all of which rely on the ability to control infections that occur in the context of compromised tissue barriers and immunologic defenses. The CDC estimates that in 2022 alone, more than 235 million antibiotic prescriptions were dispensed from pharmacies in the US, representing an annual rate of ~7 prescriptions per 10 Americans[1]. While antibiotics limit bacterial replication and spread, the host immune system plays a fundamental role clearing remnants of infection[2,3]. There are reports of different antibiotic treatments

impacting the host immune response to the underlying infection in distinct ways[4,5]. Some of these effects may be beneficial for patients, for example: increasing gut IL-10 production[6], and sensitizing bacteria to killing via host antimicrobial peptides[7,8]. Other effects however can be detrimental, such as increased virulence factor transcription[5], increased antibiotic resistance[9–11], diminished intestinal barrier integrity[12], and induced cross-resistance to innate immune antimicrobials[10,13]. Collectively these prior reports cover a large variety of different antibiotics, types of infecting bacteria, and immune impacts. Consequently, overarching patterns have remained elusive.

There are two broad mechanistic categories of antibiotics: bactericidal (cidal) drugs that directly kill bacteria and bacteriostatic

[1]Emory University/NIAID Graduate Partnership Program, Bethesda, MD, USA. [2]Signaling Systems Section, Laboratory of Immune System Biology, NIAID, Bethesda, MD, USA. [3]Department of Laboratory Medicine, NIH Clinical Center, NIH, Bethesda, MD, USA. [4]Bacterial Pathogenesis and Antimicrobial Resistance Unit, Laboratory of Clinical Immunology and Microbiology, NIAID, Bethesda, MD, USA. [5]Division of Infectious Diseases, Emory University School of Medicine, Atlanta, GA, USA. [6]Emory Antibiotic Resistance Center, Atlanta, GA, USA. [7]These authors jointly supervised this work: David S. Weiss, Iain D. C. Fraser. ✉e-mail: david.weiss@emory.edu; fraseri@niaid.nih.gov

(static) drugs that inhibit bacterial growth without causing death[14,15]. Examples of cidal drug classes we assessed include beta-lactams (ampicillin, ceftriaxone, and meropenem) that inhibit cell wall synthesis by binding different penicillin binding protein (PBP) enzymes[16,17] and quinolones (ciprofloxacin) that disrupt DNA synthesis[18,19]. Examples of static drugs we assessed include tetracyclines (tetracycline and doxycycline) that bind the bacterial ribosomal 30S subunit[20,21] chloramphenicol which inhibits the bacterial ribosomal 50S subunit[22,23], and nitrofurantoin which targets particular classes of bacterial mRNA to ultimately inhibit pathogen-specific protein synthesis as well as several other important bacterial processes[24,25]. Although nitrofurantoin can be cidal in some clinical settings because of its unique pharmacology, it is primarily a static acting drug as used in laboratory settings[26,27]. A comprehensive meta-analysis found no intrinsic superiority of bactericidal compared to bacteriostatic agents when prescribed appropriately[28]. However, these two classes of treatment present very different scenarios to the immune system. The result of cidal antibiotic treatments is damaged, non-viable bacterial cells and solubilized cellular components. In contrast, static antibiotic treatments yield viable, growth-arrested bacteria. These types of treatments thus generate two conceptually distinct sets of immunomodulatory stimuli. How this translates into actual downstream host innate immune responses to infections treated with each type of antibiotic is not known[4,5]. As a result, clinicians rarely account for immunologic impacts of their antibiotic treatments.

Macrophages use a wide variety of pattern recognition receptors (PRR) to respond to infections[29,30]. Previous work has identified the Toll-Like Receptor (TLR) pathways as important sensor systems in macrophages and other innate immune cells[31]. TLRs detect defined pathogen associated molecular pattern (PAMP) ligands that occur in bacteria including LPS (recognized by TLR4)[32], bacterial lipoproteins (recognized by TLR1 and TLR6)[31], lipoproteins and peptidoglycan (recognized by TLR2)[33,34], flagellin (recognized by TLR5)[35], and endocytosed bacterial DNA (recognized by TLR9 in the endosome)[36]. When TLRs encounter their ligands they recruit the adaptors MyD88 and/or TRIF, which in turn recruit additional factors into signaling complexes that ultimately allow transcription factors from the NF-kB, AP-1 and IRF families to upregulate a variety of key pro-inflammatory cytokines[37,38]. Different mixtures of PAMP ligands are known to influence macrophage signaling[39–41]. We hypothesized that treatment of bacteria with different types of antibiotics (namely cidal antibiotics that kill the bacteria vs. static antibiotics that do not) would impact the repertoire of PAMPs presented to immune cells, and thus subsequent immune responses. These differences could result in important, currently underappreciated[5], differences in immune control of treated infections, that could profoundly influence eventual treatment outcomes.

In this study we assessed how different classes of antibiotic treatments impact host innate inflammatory responses to infection. In an acute in vivo peritonitis mouse model we observed a static drug treatment to be far more protective (and less inflammatory) than its cidal counterpart. This indicates that, in some high bacterial load infections, cidal-driven inflammation can be severe enough to cause near complete cidal antibiotic treatment failure. We next found that cidal antibiotic treatments that kill bacteria induced greater cytokine responses from macrophages than static antibiotic treatments that halt bacterial growth. We observed this phenotype across multiple cytokine readouts, using several clinical Gram-negative bacterial isolates. Further mechanistic investigation revealed this effect to be dependent on TLR9 sensing of bacterial DNA liberated specifically by cidal but not static antibiotics. The enhanced macrophage response to cidal-treated bacteria compared to static-treated bacteria was abrogated either in the presence of DNase or the absence of TLR9. Finally, in contrast to wildtype (WT) mice, TLR9 deficient infected mice treated with a cidal drug survived at the same high rates as those treated with a static drug, and systemic inflammatory cytokine levels were equalized. Overall, these data establish a new link between how much DNA a particular antibiotic causes bacteria to release, subsequent TLR9-driven macrophage inflammation (or lack thereof), and survival outcomes in a murine peritonitis model.

## Results

### Stark survival disparity between cidal treatments and static treatments in an in vivo peritonitis model

Bactericidal (cidal) drugs directly kill bacteria, whereas bacteriostatic (static) drugs arrest bacterial growth. We sought to determine if this fundamental difference between antibiotic types might drive different responses to infections treated with these different classes of bacteria. We first selected a clinical *E. coli* (cEC1) isolate that had broad susceptibility to a wide range of cidal and static antibiotics. We assessed the minimum inhibitory concentration (MIC) for each drug via a standard broth microdilution assay, in which the bacteria were cultured in media containing serial dilutions of each drug to determine the lowest concentration that inhibited visible growth at 18 h (Table 1). The four cidal drugs in the panel were meropenem (mero), ciprofloxacin (cipro), ceftriaxone (ceft), and ampicillin (amp). The four static drugs in the panel were tetracycline (tet), doxycycline (doxy), chloramphenicol (chlor), and nitrofurantoin (nitro). Although nitrofurantoin is classified by the FDA as cidal in the context of urinary tract infections due to its tendency to concentrate in urine[42], in this study we used only lower concentrations where we observed growth arrest but no killing. MIC breakpoints were assigned according to the current Clinical & Laboratory Standards Institute (CLSI) definitions[43]. Both cEC1 and K12 (a widely used laboratory *E. coli* strain) were susceptible to all eight drugs (Table 1).

We quantified bacterial colony forming units (CFU) at a range of concentrations and time points to determine antibiotic concentrations and exposure intervals at which each cidal drug killed the bacteria, and each static drug arrested growth without killing (Fig. S1A, B). All bacteria were inoculated at $10^6$ bacteria/mL in DMEM media, and by 6 h concentrations of all cidal drugs in the 5–10× MIC range had killed all

## Table 1 | Drug information and Strain Specific MIC values

| Drug | K12 (ug/ml) | cEC1 (ug/ml) | Drug Type | Drug Mechanism |
|------|-------------|--------------|-----------|----------------|
| Mero | 1.00 | 0.03 | Cidal | Disrupts bacterial cell wall synthesis. |
| Cipro | 0.50 | 0.03 | Cidal | Inhibits bacterial DNA replication. |
| Ceft | 0.03 | 0.04 | Cidal | Disrupts bacterial cell wall synthesis. |
| Amp | 24.89 | 16.00 | Cidal | Disrupts bacterial cell wall synthesis. |
| Tet | 2.44 | 0.72 | Static | Inhibits bacterial protein synthesis (binds 30S subunit). |
| Doxy | 1.92 | 0.96 | Static | Inhibits bacterial protein synthesis (binds 30S subunit). |
| Chlor | 2.38 | 2.61 | Static | Inhibits bacterial protein synthesis (binds 50S subunit). |
| Nitro | 7.68 | 32.00 | Static | Inhibits bacterial citric acid cycle, as well as DNA/RNA synthesis. |

Strain Specific MIC Values. MIC values for each antibiotic quantified by broth microdilution in ug/mL for K12 and cEC1 strains. Values reported are the averages across three measurements across three independent experiments.

the bacteria (Fig. S1A, B), while all static drugs had halted growth without killing in the 1-5x MIC range (Fig. S1A, B). We used this MIC-adjusted approach to select antibiotic concentrations for all further experiments.

To determine how different types of antibiotic treatments impact infection pathogenesis outcomes we developed an in vivo peritonitis infection model comparing cidal-treated (cipro), static-treated (tet), and untreated mice. We infected WT mice with equal, high doses ($10^9$ CFU) of live cEC1 bacteria via intraperitoneal (IP) inoculation, then treated 30 min later with a single dose of antibiotic (Fig. 1A). Survival was then assessed over the next 30 h (Fig. 1B). We observed a large, statistically significant difference in survival between the two antibiotic treatment groups: the ciprofloxacin (cidal) treated mice had an 11.1% survival rate, whereas tetracycline treatment (static) was 77.8% protective (Fig. 1B). Additionally, we measured bacterial burdens in a peritoneal lavage, the left lobe of the liver, and the spleen, 4 h post infection in this model (Fig. 1C). Both drugs reduced bacterial burden (by ~2-4 log-fold) in all three anatomical sites compared to pre-treatment input levels as well as bacterial burdens in untreated control mice, leading to similar burdens at the later time points in both antibiotic treated groups. The difference in survival rates we observed after cidal vs. static treatments could, therefore, not be accounted for by any intrinsic difference in bacterial load or the in vivo antibacterial efficacy of these two antibiotics. Together these results highlight the importance of antibiotic selection in influencing the survival outcome of these infections, and suggest that there are mechanisms beyond differential bacterial control accounting for the large survival difference we observe between cidal and static treated groups.

To assess if differential host inflammatory responses could be an important mechanism driving these survival differences, we next measured representative proinflammatory serum cytokine levels in WT mice 1 and 2 h after they received equivalent amounts of either cidal antibiotic-treated cEC1 *E. coli* bacteria or static treated cEC1 *E. coli* bacteria via IP injection (Fig. 1D). To assess the efficacy of these antibiotic treatments, we quantified the inocula prior to injection: this quantification verified that the treated bacteria were either completely killed (cidal) or completely growth arrested (static) (Fig. 1E). The mice that received cidal (cipro) treated cEC1 had statistically significantly higher serum levels of TNF, IL12p40, and CCL3 at 1 h post infection than the mice that received static (tet) treated cEC1 (Fig. 1F). By 2 h, the mice that received cidal treated bacteria had significantly elevated TNF, IL6, IL12p40, CCL3, and CCL5 as compared to the mice that received static treated bacteria (Fig. 1F). These results are consistent with the survival data (Fig. 1B), and show that different types of antibiotic treatments that ultimately have roughly equivalent effects on bacterial clearance (Fig. 1C) can cause different degrees of detrimental inflammation in vivo.

## Bacteria treated with bactericidal antibiotics induce more inflammatory cytokines from macrophages than bacteriostatic drug treated bacteria

To further characterize this increased cidal-mediated inflammation (relative to static drugs), we infected RAW264.7 (RAW) macrophages with K12 *E. coli* at a variety of multiplicities of infection (MOIs) and treated with a range of concentrations of several cidal and static antibiotics (clustered around the effective dose ranges identified in Fig. S1A, B) for up to 8 h. The RAW macrophages infected with cidal-treated bacteria secreted more tumor necrosis factor alpha (TNF) – an important macrophage pro-inflammatory cytokine – than untreated bacteria, in a dose dependent manner with increasing drug concentrations (Fig. 2A). By contrast, macrophages infected with static-treated bacteria produced less TNF as the drug concentration increased (Fig. 1B). TNF levels increased in relation to MOI for both antibiotic classes (Fig. 2A, B). This pattern extended to several static

(Fig. S2A) and cidal (Fig. S2B) drugs in the panel. To facilitate further direct comparisons, we selected an MOI of 10, as this condition generated the greatest TNF induction. We also assessed host cell viability over a time course of infection with a small panel of clinical isolates and determined that 6.5 h was optimal for balancing cytokine signal detection while minimizing any host cell death due to infection (Fig. S3A–C). We then challenged immortalized bone marrow-derived macrophages (iBMDMs) with the clinical *E. coli* 1 (cEC1) isolate treated with each individual drug in the panel using identical drug dosing at levels that preserved cidal/static functionality (5x MIC), and Kdo2-Lipid A (KLA, the active moiety of LPS) as a positive control. We observed that cidal-treated bacteria induced ~23-271% more TNF than the static-treated bacteria, with a 118% increase on average (Fig. 2C).

To determine whether bacterial death was a primary requirement for the cidal-mediated enhanced inflammatory cytokine output we observed, we took advantage of the ability of static antibiotics to kill bacteria at high concentrations[44,45]. We infected iBMDMs with cEC1 treated with tetracycline over a concentration range likely to cover bacterial death-inducing doses based on initial optimizing experiments. This was confirmed through direct assessment of bacterial killing (Fig. 2D). As the tetracycline concentration increased, we observed two distinct patterns; first, bacterial killing increased in direct proportion to tetracycline dosage (Fig. 2D, grey bars). Second, WT iBMDMs infected with tetracycline-treated bacteria produced increasingly more TNF consistent with increased bacterial damage and death (Fig. 2D, black bars). Thus, we demonstrate that different concentrations of the same antibiotic can lead to no inflammatory enhancement at low bacterial growth-halting levels, while inducing a strong macrophage inflammatory response at higher bacterial damage-inducing levels. Further, these results suggest that the inflammatory impact of a particular antibiotic may be predicted from its degree of bacterial killing.

## Pattern of bactericidal drug mediated inflammatory enhancement is preserved in several clinical strains

Given the cidal antibiotic-induced increases in cytokine production from the lab K12 and clinical cEC1 strains, we next asked if this effect was consistent across a larger Gram-negative strain panel. We collected 10 Gram-negative patient isolates (*E. coli* (*n* = 4), *Enterobacter* (*n* = 4), and *Klebsiella* (*n* = 2)) and performed antimicrobial susceptibility testing by broth microdilution to determine MICs for the antibiotics included in the test panel. We then infected macrophages with this expanded panel of clinical isolates at a range of MOIs. Although we observed some strain-specific variability in the overall kinetics of the TNF responses to the different bacteria, when we average cidal-treated bacteria (across 3 different drugs: meropenem, ciprofloxacin, and ampicillin) and static-treated bacteria (across 3 different drugs: tetracycline, doxycycline, and chloramphenicol) at nearly every MOI measured, in nearly every strain measured, cidal-treated bacteria induced significantly more TNF than comparable amounts of static-treated bacteria (Fig. 2E).

We extended this analysis to screen a large multiplex panel covering a broad range of inflammatory cytokines and chemokines that are known to be released by macrophages at the beginning of an antibacterial innate immune response. The resulting pattern was similar to the TNF results described above: macrophages infected with cidal treated bacteria produced more potent cytokine responses than those infected with static treated bacteria using 4 *E. coli* (Fig. S4A), 4 *E. cloacae* (Fig. S4B) and 2 *K. pneumoniae* clinical isolates (Fig. S4C). The highest magnitude secreted cytokines included classically inflammatory cytokines TNF and IL6, and chemokines CCL5 and CXCL1 (Fig. S4A–C), reinforcing our choice of TNF as a representative cytokine. Additionally, we observed the cidal > static pattern for other cytokines and chemokines, although the overall

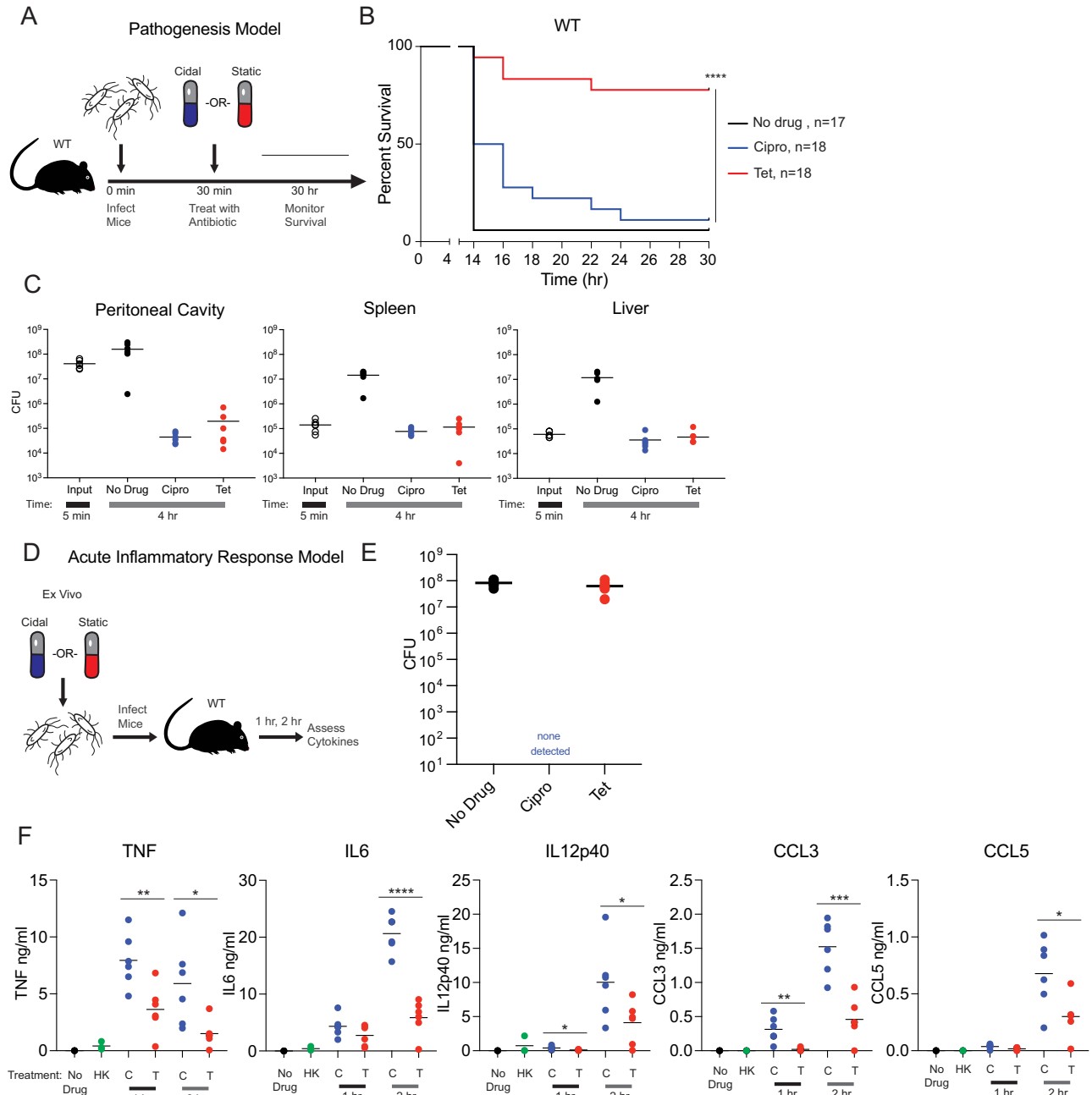

**Fig. 1 | Class of antibiotic influences host survival and cytokine responses in an in vivo peritonitis model. A** Schematic representation of peritonitis survival experiments. Mice were challenged with $10^9$ bacteria via IP injection, given the indicated antibiotic 30 min later, and survival was monitored out to 30 h post infection. **B** Survival is quantified over 30 h in the indicated number of infected WT mice (displayed as n on the graphs): 17 untreated mice and 18 mice/antibiotic treatment group. Data are pooled across three independent experiments and shown in full. Statistics shown are Kaplan-Meier survival tests comparing survival of cidal treated vs. static treated mice in each genotype. NS is non-significant, and ****$P < 0.0001$. **C** Bacterial burdens were quantified by colony forming unit (CFU) plating from a wash of the peritoneal lavage (PL), and homogenates of the spleen and the left lobe of the liver at 5 min post infection (before antibiotic treatments), and after 4 h after the indicated treatments. Data are pooled from groups of 6 mice (3 male, 3 female) across two independent experiments and shown in full. **D** Schematic representation of ex vivo cytokine quantification experiment. cEC1 bacteria were treated until either completely killed (cidal drugs)

or growth halted (static drugs), then used in identical quantities to infect mice. Cytokines were quantified 1 and 2 h later. **E** Plating of ex vivo inocula prior to beginning infections. Six samples were collected and dilution plated from each culture; data shown is representative of three independent experiments. **F** Mice were challenged via IP injection with vehicle alone (negative control, black), heat killed bacteria (green), cidal-killed bacteria (ciprofloxacin, blue), and static growth limited bacteria (tetracycline, red). Each mouse received $10^6$ CFU of bacteria in a 100 ul injection. Cidal killed bacteria were treated with drug ON to ensure complete killing, whereas static growth halted bacteria were dosed 3 h prior to infection. Mice were euthanized at 2 h post infection, blood was collected via cardiac puncture, spun down to serum, and promptly frozen. Indicated cytokines were quantified via BD cytokine bead array from serum samples. NS is $P > 0.05$, *$P < 0.05$, **$P < 0.01$, ***$P < 0.001$, ****$P < 0.0001$ by two-tailed $t$ test. Representative data from six mice/group is shown from two-four independent experiments. Source data and exact $p$-values log rank test values are provided as a Source Data file.

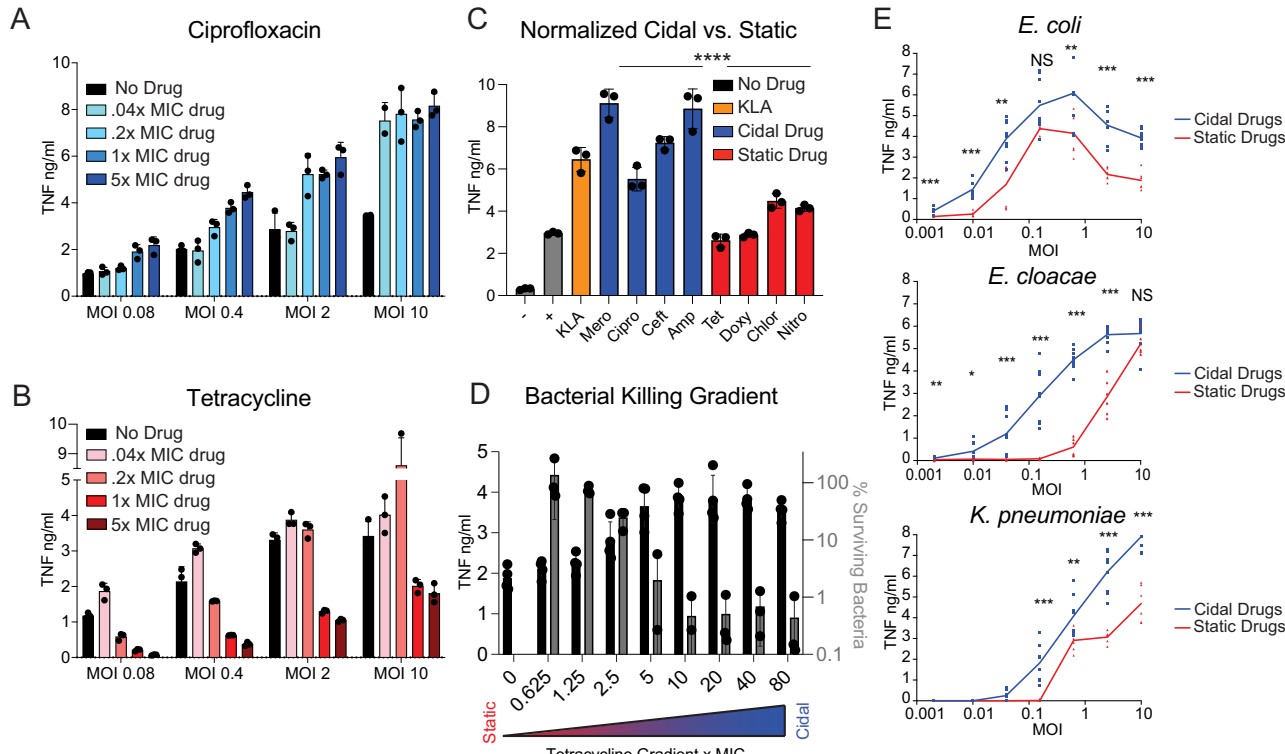

**Fig. 2 | Cidal treated bacteria Induce more TNF from macrophages than static treated bacteria. A**, **B** TNF quantified by ELISA at 8 h from RAW macrophages infected with drug treated K12 *E. coli* at a range of MOIs (moieties of infection), and antibiotic concentrations. Error bars display SEM. **C** TNF quantified by ELISA at 6.5 h from WT iBMDMs infected with 5xMIC drug treated bacteria across all the antibiotics. − is media only, + is bacteria without antibiotics, KLA is Kdo2-Lipid A. Error bars display SEM. NS is $P > 0.05$, *$P < 0.05$, **$P < 0.01$, ***$P < 0.001$, ****$P < 0.0001$, by two-tailed $t$ test; statistical comparisons in 2 C are of cidal drugs (as a group) to static drugs (as a group). We compute the average in percent increase as: (the average of all the cidals / the average of all the statics) − 1. To identify the reported range of percent increases, we computed the minimum difference (lowest cidal,

ciprofloxacin, / highest static, chloramphenicol) − 1, and the maximum difference (highest cidal, meropenem / lowest static, doxycycline) − 1. **D** Black: TNF quantified by ELISA at 6.5 h from infected macrophages at a variety of tetracycline concentrations. Grey: CFU quantification of surviving bacteria at each tetracycline concentration. Error bars display SEM. The bacterial lethality of tetracycline gradient is indicated schematically below. **E** TNF quantified by ELISA at 6.5 h from macrophages infected at various MOIs with 3 representative clinically derived strains treated with 3 cidal and 3 static antibiotics. *$P < 0.05$, **$P < 0.01$, ***$P < 0.001$, ****$P < 0.0001$ by two-tailed $t$ test. **A**–**E** display three independent measurements/group and are representative of three independent experiments. Source and exact $p$-values data are provided as a Source Data file.

magnitude was lower, including IFNβ, IL10, CCL2, and CCL7 (Fig. S4A–C).

We further expanded the screen to include several additional MOIs, and to evaluate the effect in each case we calculated the difference between cidal treated bacteria (on average) and static treated bacteria (on average) and normalized by dividing by the static average (Fig. S5A). Across 10 bacterial strains, 8 cytokine readouts, and 4 MOIs we tested 320 unique parameters, and found that cidal treatments induced more cytokine, on average, than static treatments in 299/320 instances (Fig. S5A, B). Collectively these data indicate a broad phenotype whereby Gram-negative bacteria that have been treated with cidal drugs are more inflammatory than bacteria that have been treated with static drugs.

**Small soluble factors liberated from antibiotic treated bacteria are insufficient to drive the differential inflammatory response**
Having determined that infection with cidal-treated bacteria induces higher levels of inflammatory cytokines from macrophages compared to static-treated bacteria (Fig. 3A), we sought to investigate the mechanism driving this outcome. We hypothesized that the cidal drug treatments may be liberating inflammatory PAMPs from the bacteria that the static drugs do not, thus driving the additional inflammatory signaling we consistently observed. We further hypothesized that any such signaling would occur through the Toll Like Receptor (TLR) pathway: an important innate immune sensor system that is well

known to translate initial sensing of various bacterial ligands into inflammatory cytokine output.

To test these hypotheses, we infected a variety of iBMDM cell lines that lacked individual TLR sensor components and assessed their TNF production under different infection conditions. To determine if the antibiotics alone (in the absence of bacteria) had any effect on iBMDM TNF production, we added each antibiotic to WT iBMDMs +/− KLA, and quantified TNF after 6.5 h. We did not find any increases in TNF from any antibiotic alone or in combination with KLA (Fig. S6A). Mirroring the analysis in Fig. 2C, we compared the effects of cidal treatments as a group (combined from infecting with each cidal drug) to the effects of static treatments as a group (combined from infecting with each static drug). Specifically, all cell lines were challenged with a media-only negative control, KLA treated and untreated bacteria positive controls, and the various antibiotic treated bacteria. *MyD88/Ticam-1−/−* (MyD88/TRIF) iBMDMs, in which there is no functional TLR pathway signaling, produced little to no detectable TNF in response to any of the conditions tested (Fig. 3B), demonstrating that TLR responses are required to support the increased inflammatory response we observed with cidal treated bacteria. To determine whether *MyD88/Ticam-1−/−* iBMDM cells are capable of responding to (non-TLR) stimuli, we assessed pSTAT1 protein expression in response to a 30-minute interferon beta stimulation. Both WT and *MyD88/Ticam-1−/−* iBMDM cells had comparable responses to interferon stimulation (Fig. S6B).We next infected iBMDMs lacking individual TLRs which are

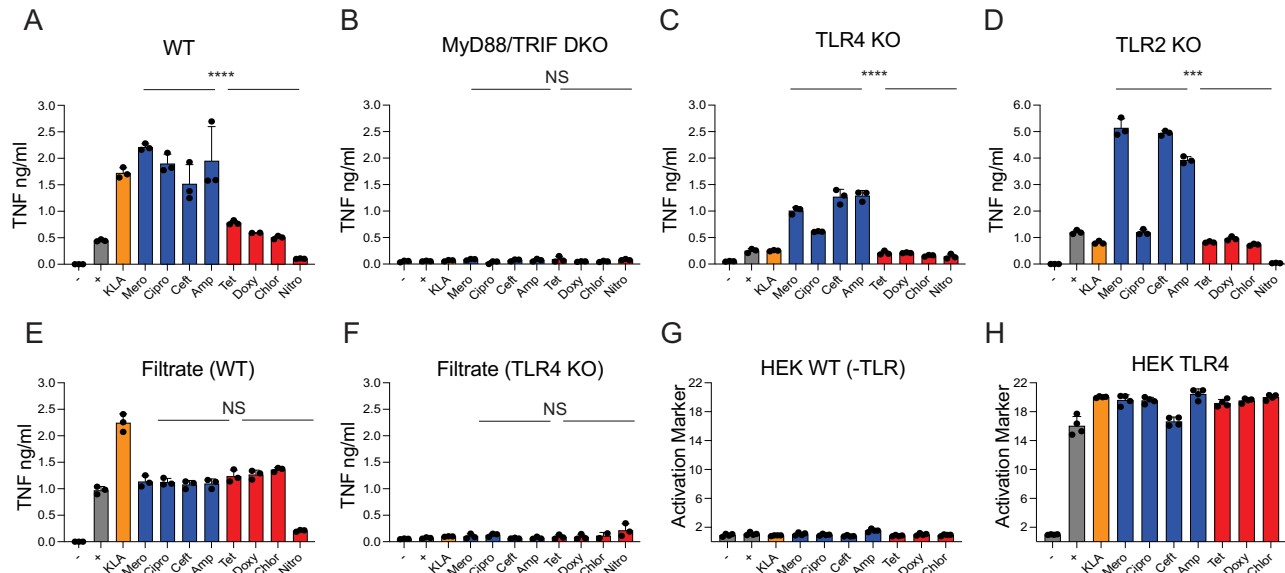

**Fig. 3 | Differential LPS release is not responsible for cidal induced increases in inflammatory cytokines.** TNF quantified by ELISA at 6.5 h from WT (**A**), *MyD88/Ticam-1−/−* (MyD88/TRIF) (**B**), *Tlr4−/−* (**C**), and *Tlr2−/−* (**D**) iBMDM macrophages infected with media alone (black), bacteria without antibiotics (black), KLA (orange), and equivalent concentrations of cidal (blue) and static treated bacteria (red) as indicated. **E**, **F** TNF quantified by ELISA at 6.5 h from WT (**E**) or *TLR4−/−* macrophages infected with only filtrate from the media alone, bacteria without antibiotics, KLA and drug treated bacteria. **G**, **H** Quanti-blue absorbance read out at

6.5 h from HEK null (**G**) or HEK TLR4 knock in (**H**) cells infected with media only, bacteria without antibiotics, KLA, and drug treated bacteria. **A–F** are representative of three independent experiments. **G**, **H** are representative of two independent experiments; all display three independent measurements/group. **A–H** *$P < 0.05$, **$P < 0.01$, ***$P < 0.001$, ****$P < 0.0001$ by two-tailed *t* test; all statistical comparisons are of cidal drugs (as a group) to static drugs (as a group). Error bars in all panels display SEM. Source data and exact *p* values are provided as a Source Data file.

known to recognize components from Gram-negative bacteria: *Tlr4−/−* (LPS) and *Tlr2−/−* (bacterial lipoproteins (BLPs)/peptidoglycan). The cidal treated bacteria induced more cytokine than static treated bacteria in these individual TLR KOs (Fig. 3C-D), indicating that differential release of these individual PAMPs is unlikely to be responsible for the broad trend of cidal drug enhanced inflammatory response we observe. There is however some variability in this trend with respect to individual drugs – notably, in the absence of TLR2 macrophages lost some response to specifically cipro treated infections (Fig. 3C), possibly indicating that cipro treated bacteria natively signal through multiple TLR pathways, and may release more BLPs/peptidoglycan than other cidal treatments.

We then sought to determine if one or more released soluble factors associated with cidal treatment of bacteria was responsible for the observed induction of cytokines. To test this, we set up the same infection conditions (positive and negative controls and bacteria +/− antibiotics), filtered the resulting solutions through a 0.22 μm filter, and treated macrophages with the flow through. The induced levels of TNF decreased to a lower level and were similar for cidal and static treated bacterial samples, and the untreated positive control (Fig. 3E). This filter experiment suggests a trend of similar levels of independently inflammatory soluble factors across the different cidal and static treatments, with a decrease in those levels for nitro-treated bacteria. Furthermore, we observed no TNF responses to any of the filtrates in *Tlr4−/−* iBMDMs (Fig. 3F). This identifies LPS as the predominant soluble inflammatory factor in the filtrates across the different antibiotic treatments.

If LPS were the common inflammatory factor driving equivalent TNF responses in the filtrates, then each drug treated group should release LPS at similar levels. To test this hypothesis, we challenged TLR4 reporter HEK cells with untreated bacteria, KLA alone, and the different antibiotic treated bacteria and measured their response via the Quanti-Blue reporter assay. We observed very similar levels of reporter activation across the entire panel (Fig. 3G, H). To determine whether these equal levels were a result of assay saturation, we also

diluted these supernatants and quantified the Quanti-blue reporter response to the diluted supernatants (Fig. S6C–F). We observed corresponding reductions of the responses (Fig. S6C–F). This suggests that none of the antibiotic treatments in our panel substantially alter the amount of LPS released by the antibiotic treated bacteria (in 6.5 h, the longest period tested) to be sensed by macrophages (Fig. 3G, H). Overall, we show that while the main soluble inflammatory component in these filtrates is LPS, the cidal drugs do not liberate more LPS than either static drugs or untreated bacteria.

**Bactericidal antibiotic treatments liberate bacterial DNA driving a TLR9-dependent macrophage inflammatory response**

Having determined that TLR signaling is critical for the response and ruled out differentially released LPS or BLPs/peptidoglycan as a driver of increased inflammation from cidal-treated bacteria, we next chose to test TLR9 as a sensor of DNA from endocytosed bacteria[46,47]. We found that when we infect iBMDMs lacking TLR9 with cidal treated bacteria, we observe diminished TNF signaling relative to the corresponding infections of WT iBMDMs, whereas signaling from static-treated bacteria was maintained (if anything we observe a slight increase in static-treated TNF signal in the *Tlr9−/−* iBMDMs relative to WT iBMDMs). The cidal vs. static difference we observe in WT infections, however, was substantially reduced in *Tlr9−/−* iBMDMs (Fig. 4A, B). This TLR9-dependence suggests that iBMDMs sense more endocytosed bacterial DNA when infected with cidal treated bacteria than they do when infected with static treated bacteria. In contrast, iBMDMs lacking STING (Fig. S7A), an important component of cytosolic DNA sensing, retained the cidal > static TNF response (Fig. S7B, C). This TLR9-restriction of the response suggests that the DNA driving the response is sensed in endocytic compartments. Equivalent interferon responses in WT and STING-deficient iBMDMs (Fig. S7D) confirm that our differential antibiotic phenotype is independent of STING, as interferon is the primary STING pathway readout.

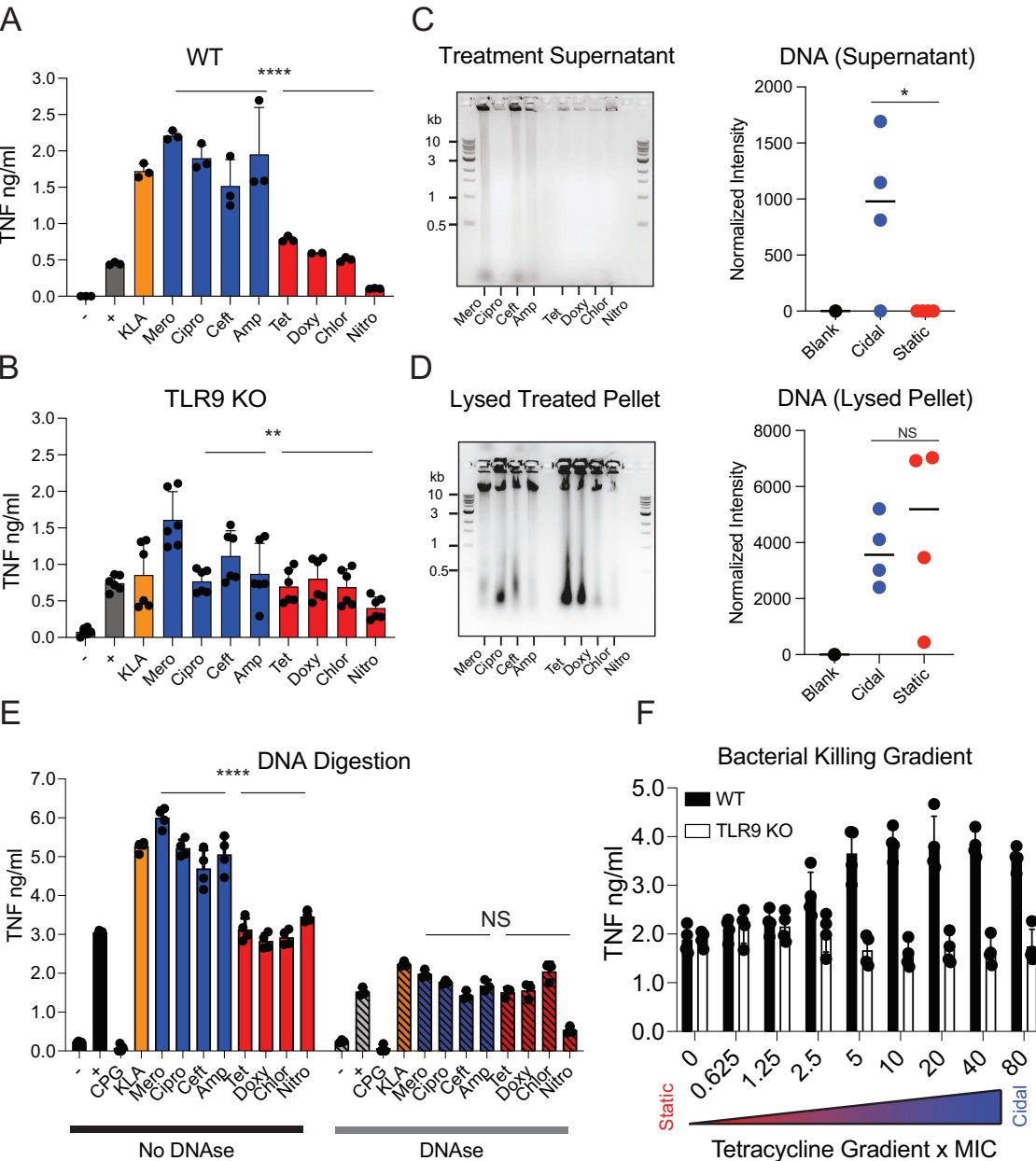

**Fig. 4 | TLR9 is required for cidal induced increases in inflammatory cytokines.** TNF quantified by ELISA at 6.5 h from WT (**A**) or *Tlr9−/−* (**B**) iBMDM macrophages infected with media alone (black), bacteria without antibiotics (black), KLA (orange), and equivalent concentrations of cidal (blue) and static treated bacteria (red) as indicated. Error bars display SEM. A is reproduced from Fig. 3A for comparison. **A** Three independent measurements/group, and (**B**) displays six independent measurements / group (aggregated across two independent replicates). **C**, **D** Bacteria were incubated with the indicated antibiotic (or no antibiotic) for 3 h, then spun down. Supernatants were collected, and the pellets of surviving bacteria were lysed. All supernatants (**C**) and pellets (**D**) were ethanol precipitated and then run on agarose electrophoresis gels. Representative ladder band sizes are labeled to the left of the gels in kilobases. Total DNA stain intensity in each lane is quantified in the graphs. **E** Infections were set up as in Fig. 3A in WT macrophages, with an

additional positive CpG DNA control, and DNaseI-XT (a nonspecific nuclease) was added to the infections where indicated. The readout is TNF quantified by ELISA at 6.5 h, with three independent measurements per group shown. Error bars display SEM. **F** TNF quantified by ELISA at 6.5 h from *Tlr9−/−* iBMDMs (white bars) infected 5xMIC drug treated bacteria across all the antibiotics. − is media only, + is bacteria without antibiotics, KLA is Kdo2-Lipid A. WT infections reproduced for comparison from Fig. 2D (black bars), lethality of tetracycline gradient is indicated below, with three independent measurements per group displayed. Error bars display SEM. **A**–**F** are representative of three independent experiments, and *$P < 0.05$, **$P < 0.01$, ***$P < 0.001$, ****$P < 0.0001$ by two-tailed *t* test; statistical comparisons in **A**, **B** and **E** are of cidal drugs (as a group) to static drugs (as a group). Source data and exact *p* values are provided as a Source Data file.

Given the TLR9 dependence we observe, we hypothesize that cidal treatments cause bacteria to release more DNA than static treatments. We tested this by determining how much DNA is released from bacteria treated with different classes of antibiotics. Bacteria were incubated +/− cidal and static antibiotics in Mueller Hinton (MH) broth media for 3 h, then supernatant and particulate fractions were run on agarose gels to assess DNA content. Nearly all the cidal

antibiotic-treated bacteria (with the exception of ciprofloxacin-treated bacteria) had visible amounts of DNA liberated into the supernatant fraction (Fig. 4C), while bacteria treated with the same dose of any static antibiotic had no visible supernatant fraction DNA (Fig. 4C). This provides direct evidence that several cidal treatments cause bacteria to release DNA into the soluble fraction, whereas static treatments do not. In contrast, the amount of DNA in the post-antibiotic particulate

fraction (surviving bacteria) trended higher in the static-treated bacteria (Fig. 4D), likely because there were far greater numbers of surviving bacteria in the static conditions. The one cidal drug that was different from the overall pattern was ciprofloxacin, the mechanism of which is to stabilize stalled topoisomerase-DNA complexes following DNA cleavage, resulting in large DNA fragments (but not bacterial cell wall disruption). This may limit our ability to identify cipro-damaged bacterial DNA in this assay, as it is likely contained within the dead, but intact, bacteria. In vivo, infected macrophages likely disassemble cipro-treated dead bacteria in their endosomes. This process may release the damaged bacterial DNA which the macrophages respond to through TLR9.

To further investigate the importance of released bacterial DNA in the cidal antibiotic-driven inflammatory response, we added a non-specific DNase to the antibiotic treated bacteria and infected WT iBMDMs. Treatment with DNase abborogated the difference between cidal and static treatments that we observed in untreated cells (Fig. 4E). This abrogation of the differential cytokine response between cidal treated and static treated bacteria reinforces the importance of released bacterial DNA to this phenotype. However, we note that the DNase itself was not well tolerated by the macrophages – by 6.5 h the DNase treated infections (including those that received no bacteria or antibiotics) had fewer healthy iBMDMs than infections without DNase. This is evidenced by cytokine responses being diminished across all DNase treated conditions compared to cells that received no DNase (Fig. 4E).

We showed previously that increased inflammation can also be induced by static antibiotics when employed at high doses that drive bacterial death (Fig. 2D). We repeated this experiment with *Tlr9*−/− iBMDMs and found that killed bacteria were unable to elicit the enhanced inflammatory output observed in WT macrophages (Fig. 4F). TNF remained at a consistent level in *Tlr9*−/− iBMDMs, independent of the static drug dose and the resulting viability of the bacteria (Fig. 4F). These data demonstrate that although TLR9 is required to support the drug-mediated increase in inflammatory output at high doses of static drug (where there is observable bacterial killing), there was also a baseline TNF component that was independent of both TLR9 signaling and the concentration of killed bacteria, likely through released LPS.

## Bactericidal antibiotics visibly liberate more bacterial DNA than bacteriostatic antibiotics

We next sought to measure bacterial DNA release under each antibiotic treatment condition in our panel. We adopted an extra-bacterial DNA-specific immune labeling strategy on Alexa 647 NHS-ester outer-membrane labeled bacterial suspensions following treatment with each antibiotic for 6 h. Extracellular DNA (eDNA) staining in the untreated and static treated bacterial groups was minimal (Fig. 5A). By contrast, we detected significant DNA release from bacteria treated with cidal antibiotics (Fig. 5A). These images suggest that much of the cidal drug treated, released DNA remained tightly associated with the dead/dying bacilli (Fig. 5A), possibly accounting for why we did not observe increased TNF responses from macrophages infected with filtered, cidal treated bacteria (Fig. 3E). We also noted bacterial clustering in the presence of external DNA in the cidal treated groups. Quantification of extracellular DNA revealed a ~ 4-10 fold increase from cidal treated bacteria compared to static treated bacteria (Fig. 5B).

To understand the kinetics of these responses we established a live-imaging platform to assess the impact of antibiotics on bacterial integrity by immobilizing antibiotic treated bacteria under a methylcellulose overlay. In this system the outer membrane of bacteria was labeled with Alexa 647 NHS-ester while the bacterial DNA was labeled with Hoechst stain. Untreated bacteria replicated and over time diluted out the NHS-ester outer membrane label (Fig. S8A, Supplementary Movie 1). By contrast, treatment with cidal antibiotics resulted in a rapid augmentation of Hoechst staining and visible leakage of DNA

resulting from compromised bacillar integrity (Fig. S8B, Supplementary Movie 2). Bacteria treated with static antibiotics failed to replicate and retained the membrane label and low Hoechst signal (Fig. S8C, Supplementary Movie S3). These data demonstrate the destructive impact that cidal drugs have on bacterial structure, and directly link cidal antibiotic treatments with increased bacterial DNA liberation.

## Bactericidal antibiotic treatment drives detrimental, TLR9-dependent inflammation in an in vivo peritonitis model

Considering our in vitro observation that cidal treatments induce DNA release that activates TLR9 in macrophages, we hypothesized that differential TLR9 signaling was the basis for the large survival differential we observe between cidal-treated and static-treated peritonitis infected WT mice (Fig. 1B). We thus conducted a similar experiment in *Tlr9*−/− mice: all mice were infected via IP injection with $10^9$ CFU bacteria, then either left untreated, ciprofloxacin treated, or tetracycline treated (Fig. 6A). In WT infected mice this resulted in near complete treatment failure in the ciprofloxacin (cidal) treatment group, and robust protection in the tetracycline (static) treatment group (Fig. 1B). In the *Tlr9*−/− mice, as the in WT mice, 93.7% of the untreated *Tlr9*−/− mice died, and 83.3% the tetracycline treated *Tlr9*−/− mice survived (Fig. 6B). However, the ciprofloxacin treated mice had far higher survival rates in the absence of TLR9: 72.2% survived to the end of the time course, as compared to 11.1% of ciprofloxacin treated WT mice (Fig. 6B, C). Bacterial burdens were assessed after 4 h in the *Tlr9*−/− mice, as with WT mice, and found to be similarly reduced by both drugs (Fig. 6C). These data directly confirm the efficacy of the ciprofloxacin treatment, and demonstrate that cidal drug treated bacteria are driving additional inflammation through TLR9 in vivo. Further, they suggest that – in this infection model – much of the antibiotic class mediated difference in survival that we observe (which is large enough to substantially change the overall survival rate) is mediated by the antibiotics' differing ability to cause bacterial breakdown and TLR9-stimulating DNA release.

Finally, we assessed proinflammatory serum cytokine levels in *Tlr9*−/− mice after they received equal quantities of either completely killed cidal antibiotic-treated cEC1 *E. coli* bacteria or entirely growth halted static antibiotic-treated cEC1 *E. coli* bacteria via intraperitoneal (IP) injection (Fig. 6D-E). WT infected mice showed substantial differences across a panel of pro-inflammatory serum cytokines when mice received the different types of antibiotic-treated bacteria (Fig. 1F). In contrast, we see no statistically significant differences in the serum cytokine levels of any of these cytokines between *Tlr9*−/− mice that received cidal-treated bacteria and those that received static-treated bacteria (Fig. 6F). These results are consistent with the lack of an antibiotic class mediated survival difference we observe in *Tlr9*−/− infected mice (Fig. 6B), and show that different types of antibiotic treatments can cause different degrees of systemic inflammation in vivo because of their differential ability to liberate bacterial DNA.

## Discussion

In this study we focused on an understudied potential driver of antibiotic treated infection outcomes: the host's innate immune response to the antibiotic treated bacteria. We found that host responses diverged strikingly depending on the mechanism of action of the antibiotic. Specifically, the β-lactam and fluroquinolone cidal antibiotics we tested that kill the bacteria induced markedly more inflammatory cytokines from host macrophages than treatment of identical infections with static antibiotics that do not kill the bacteria. This cidal-enhancing immune effect was present across different inflammatory markers, Gram negative bacterial isolates, and serum cytokines. In our in vivo peritonitis model, the increased inflammation induced by cidal drug treatment mediated lethal peritonitis in vivo, whereas static treatment did not This increased inflammation was also driven by bacterial death independent of antibiotic identity, and

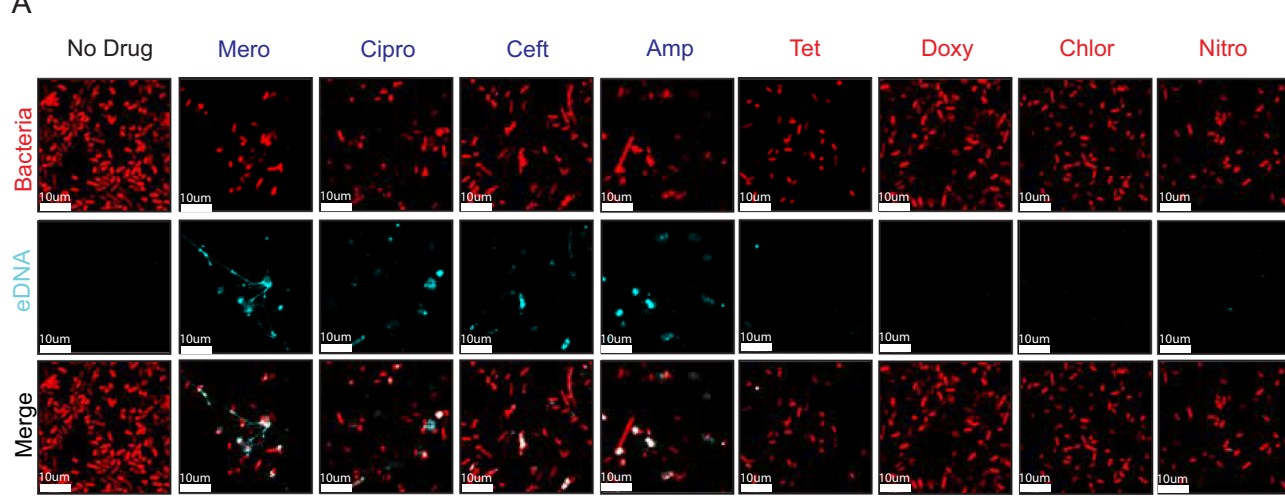

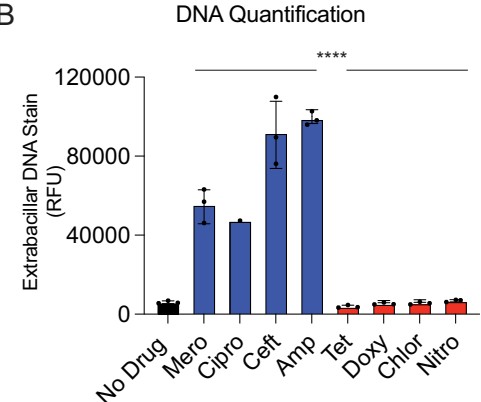

**Fig. 5 | Cidal antibiotics liberate more DNA from bacteria than static drugs.**
**A** Alexa 647-labeled cEC1 bacteria (red) were incubated with no antibiotics, static antibiotics or cidal antibiotics as indicated for 6 h before Hoechst staining for external DNA (cyan). One representative image is shown. **B** High-content imaging quantification of extracellular DNA from the samples from A-B. ***$P < 0.001$, by two-tailed $t$ test, with three independent measurements per group displayed (**A**, **B**) are representative of two independent experiments. Error bars display SEM. Source data and exact $p$-values are provided as a Source Data file.

primary mode of action. When we used artificially high doses of the normally static drug tetracycline to induce bacterial death, we saw increased TNF from infected macrophages that directly tracked the percentage of killed bacteria.

In considering the mechanism of action driving the increased cytokine responses from cidal-treated antibiotics, we initially hypothesized that cidal treatments would increase LPS release from the dead bacteria, which macrophages would sense through TLR4. Some early reports observe increased LPS shedding with various cidal drugs[48,49], though this is also been seen in bacteria naturally replicating[50], and occasionally even in static drug treated bacteria as well[51]. Surprisingly, the increased cidal-dependent inflammatory response was not driven by LPS. The cidal treated bacteria tested here still induced increased responses from TLR4 deficient macrophages, and further testing demonstrated roughly equal levels of released LPS across untreated (naturally replicating) bacteria and the various cidal and static treated conditions.

Loss of cidal-enhanced inflammation in TLR9-deficient macrophages pointed in a new direction: implicating bacterial DNA (rather than LPS) as the likely inflammation enhancing factor. We then found that only cidal treated bacteria release detectable amounts of DNA, and that DNase-mediated degradation of released DNA equalized the macrophage TNF response across the cidal and static treatment conditions. These data support the importance of bacterial DNA to the cidal-mediated inflammatory enhancement. This result is consistent with recent work in the Torres lab that also found bacterial DNA to be a driver of severe infection outcomes in *S. aureus* infections[52]. Finally we observed that cidal drug induced TLR9-driven inflammation was so strong in infected WT mice that it caused near complete treatment failure. This is consistent with a previous report implicating TLR9 responses in sepsis pathogenesis in the absence of antibiotics[53]. By contrast, we found static treatment was protective in vivo regardless of mouse genotype, but cidal treatment was only protective in TLR9 deficient animals. The magnitude of this phenotype emphasizes the potential therapeutic importance of understanding how a given antibiotic impacts downstream host immune responses to a high acuity, quickly treated infection (in addition to its impact on bacterial clearance). How different types of treatments impact chronic bacterial infections, or infections that take longer to identify and where treatment is initiated later during the course of infection, is both unclear and a worthwhile area of future investigation.

Another consideration is that some anti-infective drugs may themselves influence innate immune responses. For example, newer generations of tetracycline class antibiotics have been shown to dampen host inflammation directly[54–57], though the effect is somewhat contested in macrophages[58,59]. This is also consistent with the anti-inflammatory effects of the steroid treatments that some septic patients receive, with several RCTs demonstrating the efficacy of this

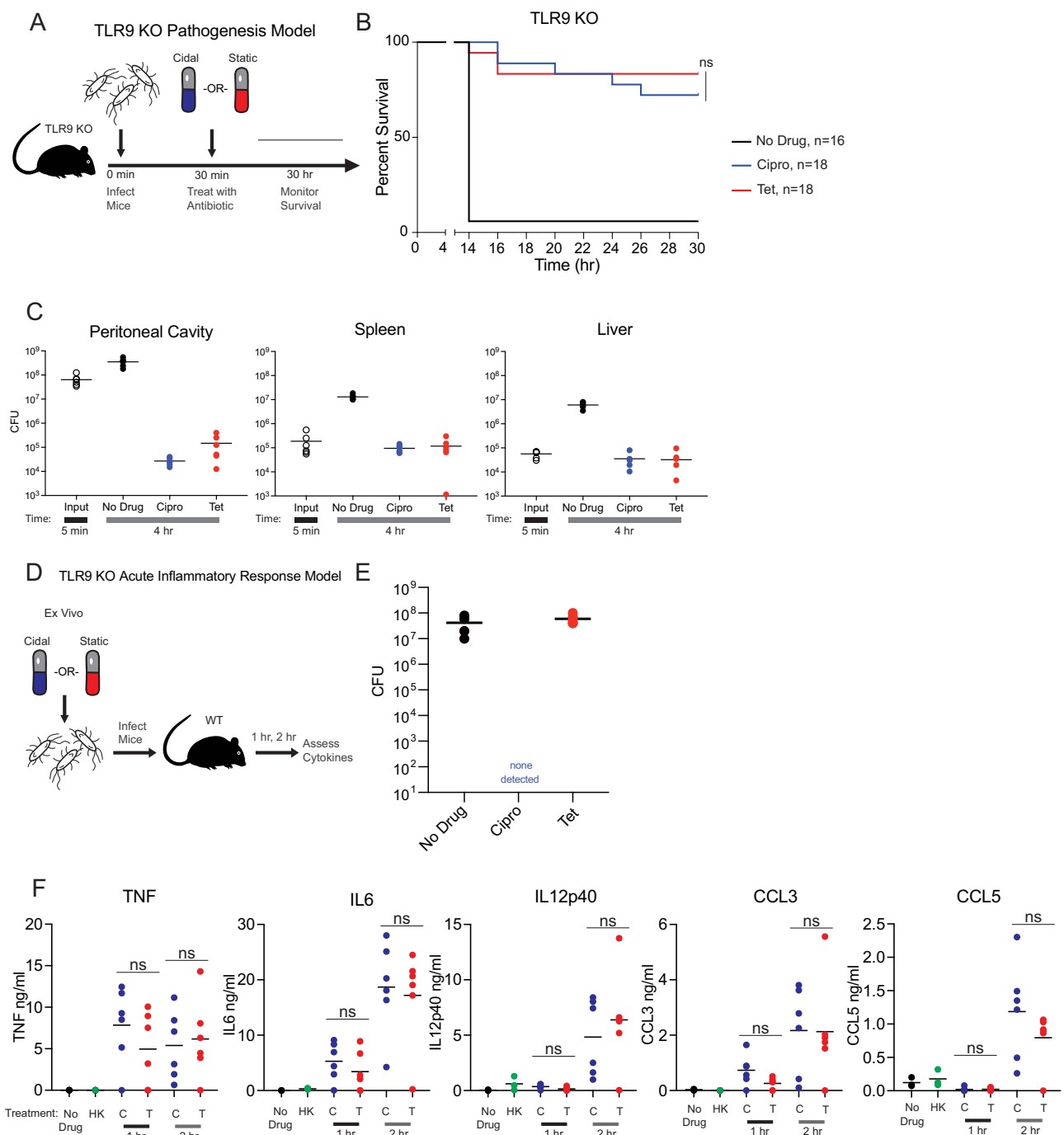

**Fig. 6 | Antibiotic class mediated survival difference depends on host TLR9 signaling. A** Schematic representation of peritonitis survival experiments. *Tlr9*−/− mice were challenged with 10⁹ bacteria via IP injection, given the indicated antibiotic 30 min later, and survival was monitored out to 30 h post infection. **B** Survival is quantified over 30 h in the indicated number of *Tlr9*−/− infected mice (displayed as n on the graphs): 16 untreated mice and 18 mice/antibiotic treatment group. Data are pooled across three independent experiments and shown in full. Statistics shown are Kaplan-Meier survival tests comparing survival of cidal treated vs. static treated mice in each genotype. NS is non-significant, and ****P < 0.0001. **C** Bacterial burdens were quantified by CFU plating from a wash of the peritoneal lavage (PL), and homogenates of the spleen and the left lobe of the liver at 5 min post infection (before antibiotic treatments), and after 4 h after the indicated treatments. Data are pooled from groups of 6 mice (3 male, 3 female) across two independent experiments and shown in full. **D** Schematic representation of ex vivo cytokine quantification experiment, reproduced for convenience from 1D. cEC1 bacteria were treated until either completely killed (cidal drugs) or growth halted

(static drugs), then used in identical quantities to infect mice. Cytokines were quantified 1 and 2 h later. **E** Plating of ex vivo inocula prior to beginning infections. Six samples were collected and dilution plated from each culture; data shown is representative of three independent experiments. **F** *Tlr9*−/− mice were challenged via IP injection with vehicle alone (negative control, black), heat killed bacteria (green), cidal-killed bacteria (ciprofloxacin, blue), and static growth limited bacteria (tetracycline, red). Each mouse received 10⁶ CFU of bacteria in a 100ul IP injection. Cidal killed bacteria were treated with drug ON to ensure complete killing, whereas static growth halted bacteria were dosed 3 h prior to infection. Mice were euthanized at 2 h post infection, blood was collected via cardiac puncture, spun down to serum, and promptly frozen. Indicated cytokines were quantified via BD cytokine bead array from serum samples. NS is P > 0.05, *P < 0.05, **P < 0.01, ***P < 0.001, ****P < 0.0001 by two-tailed *t* test. Representative data from six mice/group is shown from two-four independent experiments. Source data and exact *p*-values and log rank test values are provided as a Source Data file.

intervention[60–62]. If tetracycline class antibiotics are systemically immunosuppressive independent of their direct effects on bacteria, this highlights an advantage of using these antibiotics under conditions where enhanced inflammation could be particularly harmful. The possibility that tetracyclines may be beneficial for systemic infections both in limiting bacterial DNA release and in directly suppressing inflammation deserves further investigation, as does the possibility that pharmacological TLR9 blocking might be employed in combination with cidal drugs to minimize harmful inflammation in these types of infections. An additional example of an anti-infective drug that is known to directly influence host innate immune reactions in important ways is diethylcarbamazine (DEC) therapy for loiasis. Unlike tetracycline that can dampen host responses, DEC is known to activate host eosinophils to directly to facilitate parasite clearance[63], and it is the standard of care for definitive treatment of loa loa parasite infection in humans[64]. However, it is contraindicated for patients with high parasite loads because of known associations with potentially fatal encephalopathy complications[65–67]. Instead, physicians treat high parasite load patients with therapies that lower parasite burdens without activating host eosinophils (such as apheresis or other anti-parasitic agents that directly target parasite proteins to paralyze and kill them)[64], highlighting an additional high pathogen load infection context in which host innate responses play a critical role in determining the success or failure of the treatment.

Clinicians currently prescribe antibiotics based on bacterial susceptibility to different antibiotics. If they consider mechanism of action, they tend to preference cidal treatments despite clinical evidence that both classes are equally effective when used to treat a susceptible infection[28]. Our study introduces a new consideration: downstream immunological consequences. We have uncovered a novel link between an antibiotic's mechanism, the extent to which it causes bacterial DNA release, and its downstream impact on host innate inflammatory responses and ultimately survival (Fig. S9). We can thus envision a future where clinicians proactively consider the impact of the treated bacteria on the host immune response when prescribing antibiotics. Which inflammatory outcome is more desirable will depend primarily on the clinical context. While further study is needed in the form of randomized control trials, these data create new actionable hypotheses. For example, in cases of overwhelming inflammation, static immune limiting antibiotics may be beneficial. In immunosuppressed patients, or infections where the pathogen suppresses inflammation, perhaps cidal acting pro-inflammatory treatments may be a better choice. Increased awareness of the inflammatory impacts of specific types of antibiotic treatments could thereby allow clinicians to translate these findings into individually targeted antibiotic therapies that simultaneously control bacterial burden and modulate immune responses to best fit each patient.

## Methods

### Ethics statement

All procedures described that we performed on mice were done adhering to the Principles of Laboratory Animal Care, and pre-approved by the NIAID Institutional Animal Care and Use Committee committee (National Institutes of Health, Bethesda MD) on protocol LISB-3E. All proceedures were performed by specifically approved investigators on the relevant protocols. No human samples were used in this study. The temperature range for both inside the animal cage and the animal holding room was 20–24 °C, and the humidity range was 30–70%. A 12 h light/dark cycle (6 am–6 pm) was maintained at all times. All mice were euthanized at the conclusion of each experiment.

### Immune cells

Macrophages were maintained in culture at 37 °C, 5% humidity in Gibco Dulbecco's Modified Eagle Medium (catalog #: 11995-065) supplemented with 4.5 g/L D-glucose, L-Glutamine, 110 mg/L sodium

pyruvate and 10% BenchMark™ heat inactivated bovine fetal bovine serum from GeminiBio (catalog #: 100-106). We used the following macrophage types: RAW264.7 murine cell line from ATCC (catalog #: TIB-71), murine bone marrow derived macrophages (BMDM), and murine immortalized BMDM (iBMDM). BMDMs were isolated from WT C57BL/6 J mouse femurs and tibias and differentiated for 7 days in DMEM supplemented with 50 ng/mL MCSF. Prior to use the cells were removed from their differentiating plates and replated at 30,000 cells/well in DMEM media without growth factor in 96-well plates. iBMDMs were generously provided by Eicke Latz (University of Bonn; WT, *MyD88/TRIF−/−*, *TLR2−/−*, and *TLR4−/−*) and Kate Fitzgerald (UMass; *STING−/−*). *TLR9−/−* iBMDMs were generated from *TLR9−/−* BMDMs by infection with the J2 virus (generously provided by Howard Young, NCI, NIH), as previously described[68]. Briefly, after J2 virus infection, cells were gradually weaned off macrophage colony stimulating factor (M-CSF)-containing supportive media. When the cells grew back to appropriate levels they were expanded, and the absence of TLR9 expression, and lack of response to CpG DNA stimulation, were confirmed by flow cytometry and TNF ELISA respectively.

### Bacteria

Bacterial strains used in this study included two main model *E. coli* strains: K12 from ATCC (K12 MG1655) and the first clinical strain in our panel: clinical *E. Coli* 1 or *cEC1*. All other clinical isolates used in this study were derived from the routine care of NIH Clinical Center patients in the Department of Laboratory Medicine, were fully deidentified, and contained no residual human material. As such, their use was exempt from IRB approval. The clinical isolates included 4 *E. coli* isolates, 4 *E. cloace* isolates, and 2 *Klebsiella pneumoniae* ssp isolates. Each isolate was struck out on an MH agar plate to isolate single colonies, grown overnight (ON) from a single colony, and frozen in a 25% glycerol stock. Prior to infection, each culture was grown with shaking at 220 rpm at 37 °C ON to saturation phase in Muller Hinton (MH) broth media (Teknova, catalog #: M5899), then adjusted based on OD600 to the indicated MOI, washed 3x with PBS to clear any debris left in the media from the ON, and finally resuspended in DMEM media +/− antibiotics. To quantify colony forming units (CFU), bacteria were serially plated on MH agar plates and the initial concentration was back calculated based on colony enumeration at the first spot with colonies sparse enough to distinguish.

### Antibiotics

The standard panel of antibiotics included four cidal acting and four static acting antibiotics. The cidal antibiotics were: meropenem trihydrate (RPI, catalog #: M62500-1.0), ciprofloxacin (Sigma, catalog #: 17850-5G-F), ceftriaxone (RPI, catalog #: C54700-1.0), and ampicillin sodium salt (Fisher Scientific, catalog #: 69-52-3). The four static acting antibiotics were tetracycline HCl (RPI, catalog #: T17000-25.0), doxycycline (RPI, catalog #: D43020-25.0), chloramphenicol (RPI, catalog #: C61000-25.0), and nitrofurantoin (RPI, catalog #: N51500-25.0). Each antibiotic was dissolved from the indicated powder stock in an appropriate diluent at 10 mg/mL (or 100 mg/mL for ampicillin) and frozen. Diluents used were DMSO (meropenem, tetracycline, and nitrofurantoin), HCl (ciprofloxacin), ethanol (chloramphenicol), and endotoxin free water (ampicillin, ceftriaxone, and doxycycline). Aliquots were thawed and diluted by at least 1:100 to working concentrations into DMEM media for all in vitro experiments. All antibiotics were dosed as indicated relative to the empirically calculated, strain specific, drug specific MIC as determined by broth microdilution. To determine strain specific MICs, each strain was plated on a gradient of drug concentrations, sealed with a gas permeable seal (Diversified Biotech, catalog #: BEM-1), and incubated ON without shaking at 37 °C. 18–20 h later plates were analyzed to determine the minimum concentration of drug required to stop visible bacterial growth. The average of 6 tests conducted over two different days was

considered the MIC. A strain was considered susceptible to a given antibiotic if its MIC fell below the CLSI breakpoint for Enterobacterales as defined in their 2020 M100 guidance document. For in vivo experiments, antibiotics were prepared fresh from powder stocks. Dosing for those experiments was determined in accordance with the IACUC approved standard protocol #LISB-3E. Specifically, tetracycline was dosed at a final concentration of 40 mg/kg, and ciprofloxacin was dosed at a final concentration 50 mg/kg via IP injection in 100 uL volume 30 min after infection.

## In vitro infections

The general protocol used for in vitro infections unless noted otherwise is described here; specific alterations for particular experiments are noted in the following paragraph. Macrophages (RAW, BMDM or iBMDM as specified) were plated in antibiotic-free DMEM media on flat bottom, TC-treated 96-well plates at 30,000 cells / well and allowed to adhere ON. Infecting inoculums were prepared with bacteria at the indicated MOI and antibiotics at the indicated concentration relative to that strain's drug specific MIC in 10 mL of antibiotic free DMEM media. The negative control was media alone, and positive controls included untreated bacteria, media with 200 uM of KLA the chemically active molecule in LPS from Avanti Polar Lipids (catalog #: 699500), and when relevant media with 10 ug/mL CpG DNA from Inviogen (catalog #: tlrl-1826). Bacteria were left to incubate with the antibiotics for 45 min. Then the media was aspirated from the cell plates and replaced with infecting inoculums (or controls) and allowed to incubate for either 4 or 8 h in the K12/RAW cell initial infections, or for 6.5 h for infections of BMDMs/iBMDMs with the clinical strains. At the conclusion of the infection period supernatants were collected and frozen at −80 °C for further analysis.

For the filter experiments (Fig. 2E, H) the inoculums were prepared in bulk as described above, allowed to incubate together for 45 min, and added to macrophages alone as controls. The remainder of the inoculums were passed through a 0.22 micrometer filter and the resulting filtrate was used to infect cells. For the DNase experiments (Fig. 3G) we used DNase I-XT enzyme (New England Biosciences, catalog #: M0570L) diluted in the buffer provided by the manufacturer. The enzyme and buffer were spiked in after infecting inoculums had been added to the cells at a concentration of 2 units/well.

## ELISA

TNF was quantified from in vitro infected cell supernatants using the R&D Systems DuoSet Mouse TNF ELISA kit (catalog #: DY410) according to the manufacturer's standard protocol. In brief, Thermo Scientific NUNC MaxiSorb™ 384-well plates (catalog #: 464718) were coated in 15 ul/well of diluted capture antibody ON at 4 °C, then washed in PBS + 05% Tween-20 wash buffer and blocked in 1% BSA ON at 4 °C. Experimental supernatants were thawed, diluted by 1:4, and added to the pre-coated plate for 2 h at room temperature (RT) along with standards serially diluted down 2-fold from the TNF standard reagent in the ELISA kit. They were then removed, the plate was washed, and diluted secondary Ab was added for an additional 2 h at RT. This was removed, the plate was washed, and streptavidin horseradish peroxidase (HRP) detection reagent was added for 20 min. Finally, the plate was washed and developed using 15 ul/well of 1-Step Ultra™ TMB-ELISA substrate from Thermo Scientific (catalog #: 34028) and quenched with 15 ul/well of sulfuric acid 1-5 min later when the standard curve controls had developed to within the visible range for the assay. Plates were read on a VANTAstar plate reader at 450 nm, and results were background corrected by subtracting out noise measured at 570 nm. A standard curve was calculated using the built in VANTAstar MARS quantification software, and samples within the linear range were interpolated and corrected for the sample's initial 1:4 dilution factor.

## Luminex

iBMDMs were plated at 30,000 cells/well ON on a 96-well TC plate then infected the next day with the relevant bacterial strain at the indicated MOI in one of three conditions: untreated (control), cidal (meropenem) treated, and static (tetracycline) treated. Antibiotics were used at 5xMIC based on the strain specific calculated MIC for each drug. Supernatants were then collected and frozen for subsequent analysis using the Cytokine & Chemokine 36-Plex Mouse ProcartaPlex™ Panel 1 A (catalog #: EPX360-26092-901) kit from ThermoFisher with their additional IFNβ (catalog #: EPX01B-26044-901) and TGFβ ProcartaPlex™ (catalog #: EPX01A-20608-901) beads added in. We used the kit as per the manufacturer's protocol: magnetic beads were prepared as a master mix, added to the magnetic plates, combined with samples and pooled standards, and incubated for 2 h at RT. The plate was washed using the appropriate magnet to retain the beads, and detection antibodies were then added to the beads, and incubated for 30 min. The plate was once again washed, and the beads were resuspended in Streptavidin-PE which was incubated for an additional 30 min. The plate was washed one more time, and the samples were suspended in 100 uL of reading buffer and read on a BD Luminex™ 200 machine. Data was exported and corrected for any over-saturated standard curves, then interpolated for the various readouts based on the linear ranges of the relevant standard curves. From this panel only the following cytokines were generated above the limit of detection (LOD): TNF, CCL5, CXCL1, IL6, CCL2, CCL7, IL10, and IFNβ.

## Cytometric bead array (CBA)

To determine cytokine levels from serum samples we used the cytometric bead array (CBA) from BD Biosciences. Specifically, using the BA Mouse/Rat Soluble Protein Master Buffer Kit (catalog #: 558267) we simultaneously quantified: TNF (catalog #: 558299), IL6 (catalog #: 558301), IL12-p40 (catalog #: 560151), CCL2 (catalog #: 558342), CCL3 (catalog #: 558449), and CCL5 (catalog #: 558345) from diluted serum samples according to the manufacturer's instructions. Briefly samples and standards were diluted 1:4 in assay diluent, combined with capture beads, and incubated for 1 h at RT. Next PE Detection reagent was added and incubated at RT for an additional hour. Finally, the beads were washed, and their ApCy7 and PE fluorescence levels were determined on a BD LSR Fortessa flow cytometer. The resulting standards were mapped to BD's standard CBA coordinate system to disaggregate the individual cytokines based on their positions within the array in FloJo, and those gates were applied to each sample. Each value is the mean PE intensity of all beads measured for a given cytokine, extrapolated to a protein concentration using the corresponding standard curve.

## LPS quantification

Murine TLR4 expressing HEK293 Reporter cells from Invivogen (catalog #: hkd-mtlr4ni) were plated at 30,000 cells / well on flat bottom 96-well tissue culture plates and allowed to adhere ON at 37 °C. The following day they were infected as described with *cEC1* +/− antibiotics and with KLA as the positive control. This cell line has a secreted embryonic alkaline phosphatase (SEAP) reporter under the control of *NFkB* promoter that we used to read out the activity of the NFkB pathway in the various infections. SEAP reporter activity from the supernatants of these infections was quantified by incubating samples with Invivogen QUANTI-Blue™ reagent (catalog #: rep-qbs). Absorbance was then read out at 620 nm on a VANTAstar plate reader, as per the manufacturer's instructions.

## Western Blot

WT, MyD88/TRIF DKO, and STING KO iBMDMs were plated at 800,000 cells per well in 12-well plates, and stimulated as described in the figures. Cells were then lyzed in 1X SDS-PAGE loading buffer with protease and phosphatase inhibitor cocktails (Roche). Samples were

resolved on a 4%–20% gradient SDS-PAGE gel. Following transfer of the proteins, the nitrocellulose membrane was blocked in 5% BSA for 1 h and probed with the following antibodies (1:1000) overnight at 4 °C in 1% BSA: GAPDH, (CST#5174), Phospho-STAT1 (CST#9177S), and STING (CST#13647). Western blots were incubated with the respective HRP-conjugated secondary antibodies and visualized using ECL reagents (Pierce) and imaged on a ChemiDoc gel imager (Bio-rad).

## Released DNA characterization

Bacteria were cultured ON and prepared as previously described, incubated with various antibiotics for 3 h, and spun down for 20 min at $10,000 \times g$. The supernatants were collected, and the remaining pellets were lysed in 0.01% SDS lysis buffer. All samples were then precipitated with 1/10th of the original supernatant volume of 3 M sodium acetate and 2.5x original supernatant volumes of 100% EtOH, vortexed thoroughly, and incubated for 30 min. Samples were then spun down at $10,000 \times g$ for 20 min, and resuspended in water and NEB DNA gel loading dye (catalog #: B7024S). Samples and NEB Quick-Load 1 kb Plus ladder (catalog #: N0469S) for sizing were then run on a 1% electrophoresis gel at 100 V for 25 min and imaged on a Bio Rad ChemiDoc™ MP imaging system.

## Bacterial imaging

Bacteria were grown overnight in 5 mL Miller-Hinton Broth (MHB) followed by pelleting at $4000 \times g$ for 10 minutes. Bacteria were resuspended in 500 uL 10% sodium bicarbonate in water and re-pelleted. 100 ug Alexa 647 NHS-ester (catalog #: AAT 1833) was dissolved in 500 uL 10% sodium bicarbonate in water and was immediately used to resuspend the bacterial pellet. After 5 minutes of labeling, bacteria were pelleted at $10,000 \times g$ for 1 minute followed by 2 washes in 500 uL DMEM lacking phenol-red. For fixed imaging, 100 uL of labeled bacteria was added to 400 uL of antibiotic-containing medium. Samples were plated in 384-well format in Sigma 0.1% poly-L-lysine (catalog #: P8920) coated wells and were fixed with 4% paraformaldehyde (PFA) for 10 minutes at indicated timepoints. Fixed bacteria were washed in PBS followed by blocking in 2% BSA PBS for 30 min. External bacterial DNA was stained with mouse anti-DNA antibody (catalog #: SC-58749) at a 1:200 dilution and secondary 488-donkey anti-mouse at 1:2000. For live imaging, 647-labeled bacteria were stained with Thermo Hoechst 33342 (catalog #: H3570) at 50ug/mL in DMEM lacking phenol red, and containing the indicated antibiotics. Ten uL of bacterial suspension was layered on a Thermo Labtek II chambered cover-glass (catalog #: 155411) followed by the addition of a layer of 300 uL of 2% methylcellulose containing 1× indicated antibiotic. Live imaging was performed on a Leica SP8 confocal microscope at 4x objective magnification where images were acquired every 5 minutes for up to 6 h.

## Mice

All mice were originally obtained from Jackson Laboratories or bred in house from those initial mice. We used the following strains in this study: WT (C57BL/6 J, # 000664), and TLR9−/− (C57BL/6-$Tlr9^{em1.1Ldm}$/J, #034449 – these mice were bred to homozygous KO in house). All mice were maintained throughout the duration of our studies under specific pathogen free (SPF) conditions.

## Mouse Infection Models

For the peritonitis pathogenesis model (Figs. 1A–C and 6A–C) cEC1 E. coli was grown ON at 37 °C to stationary phase, washed 3x in PBS, and resuspended in DMEM media at a final concentration of $1^{10}$ bacteria/mL (as quantified by direct serial dilution plating). 100 ul of this preparation was injected IP to begin the infections. In the antibiotic treatment groups antibiotics were given by IP injection 30 min after the initial infecting inoculum. All mice were then monitored for signs of distress over the next 30 h, and euthanized either when they began exhibiting clinical indicators of morbidity (ruffled fur, tail drooping, shaking, unable to move when prompted, etc.) or at the conclusion of the study. Survival times were recorded and graphed on Kaplan Mayer curves using the survival graph template in GraphPad PRISM. To assess bacterial burdens: mice were euthanized at the indicated times, the peritoneum was washed with 3 mL of 4 °C PBS, and the spleen and left lobe of the liver were collected. Spleens and livers were homogenized in 1 mL of 4 °C PBS. Peritoneal washes, spleen homogenates, and liver homogenates were dilution plated on LB agar plates and incubated ON at 37 °C. Finally, CFUs were quantified from these plates.

For the ex vivo, pre-treatment acute inflammatory response model (Figs. 1D–F and 6D-F), cEC1 E. coli was grown ON at 37 °C to stationary phase, washed 3x in PBS, and resuspended in DMEM media containing static drugs (3 h prior to infection) or cidal drugs (ON prior to infection). Heat killed control bacteria were incubated at 100 °C for 30 min prior to use. Infecting inoculums were quantified to verify that the treatments functioned as intended: growth halting for static acting drugs, and complete bacterial killing for cidal acting drugs and heat killed controls. Bacteria (or media alone for the negative controls) were then administered to mice via IP injection. Mice were sacrificed at 1 and 2 hpi and blood was collected by cardiac puncture into heparin micro tubes on ice. Blood was spun at $3000 \times g$ for 20 min at 4 °C and serum was collected and promptly frozen at -80 °C for further analysis.

## Statistics & reproducibility

In vitro assays were conducted in 3–6 wells/condition as indicated in the legends, and repeated 2–6 independent times also as indicated in the legends for individual experiments. No statistical method was used to pre-determine sample sizes, no technically sound data were excluded from the analyses, these experiments were not randomized, and investigators were not blinded to allocation during experiments and assessment (because all analyses were conducted simultaneously and uniformly on entire data sets, not individual conditions). All statistics conducted were 2-tailed student's t tests, ran in Excel, on the groups indicated on the graphs and in the ledgends (generally all data from cidal drugs in a given experiment vs. all data from static drugs in a given experiment).

In vivo assays included 16–18 mice per group in the survival studies (data was collected on three separate occasions and is shown in the aggregate), 6 mice per group in the CFU quantification studies (data was collected on two separate occasions and is shown in the aggregate), and 12 mice per group in the cytokine quantification studies (data was collected on two separate occasions and is shown as a representative replicate). All experimental groups across all in vivo experiments were split evenly between male and female mice, and consisted of mice 6-9 weeks old at the time of the experiment. No statistical method was used to predetermine sample sizes, and no technically sound data was excluded. Mice were all randomized into the different groups. There was partial blinding in the survival experiments (facility technicians who were not aware of the experimental groupings made initial morbidity determinations). Otherwise, experimenters were not blinded because all data was analyzed simultaneously and uniformly across entire data sets. Survival data is displayed as Kaplan Meier curves, and differences were assessed statistically between cidal and static groups by log rank test. Individual time-adjusted cytokine quantification significance comparisons were assessed by students 2-tailed t-test between indicated groups.

## Reporting summary

Further information on research design is available in the Nature Portfolio Reporting Summary linked to this article.

## Data availability

All data are available in the main text or the supplementary materials. Source data are provided with this paper.

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

## Acknowledgements

We thank Elizabeth Fischer, Sophia Anatolia Schmitt, Margery Smelkinson, Juraj Kabat, and Tom Moyer (all at NIAID Research Technologies Branch) for providing help with initial optimization of imaging systems for individual bacteria across various platforms. We thank Kristina Lamont, Tierra Hunt, and all the NIAID animal care staff for their outstanding mouse husbandry and their technical help with mouse experiments. We thank colleagues in the Laboratory of Immune System Biology, Pam Schwartzberg, Hannah Ratner, and Victor Band for helpful discussions and critical reading of the paper. DSW is a Burroughs Wellcome Fund Investigator with a Pathogenesis of Infectious Disease award that supported this study, and was supported as well by National Institutes of Health grant AI158080. This study was supported by the Intramural Research Program of the National Institute of Allergy and Infectious Disease (NIAID), National Institutes of Health (NIH).

## Author contributions

Conceptualization: J.L.G., D.S.W., and I.D.C.F. Methodology: J.L.G., C.J.B., R.B., and J.S. Investigation: J.L.G., C.J.B, R.B., J.S., S.P.J., and S.D. Resources: S.D., J.P.D., and I.C.D.F. Data Curation, J.L.G.: D.S.W., and I.D.C.F. Writing – Original Draft: J.L.G., D.S.W., and I.D.C.F. Writing – Review & Editing: J.L.G., C.J.B., R.B., J.S., S.P.J., S.D., J.P.D., D.S.W., and I.D.C.F. Visualization: J.L.G., C.J.B. Supervision: D.S.W., and I.D.C.F. Project Administration: J.L.G., and I.D.C.F. Funding Acquisition: D.S.W. and I.D.C.F.

## Funding

## Competing interests

The authors declare no competing interests.
