## [Transparent Peer Review file · Nature Communications]

Bactericidal Antibiotic Treatment Induces Damaging Inflammation via TLR9 Sensing of Bacterial DNA

Corresponding Author: Dr Iain Fraser

Version 0:

Reviewer comments:

Reviewer #1

(Remarks to the Author)

In this manuscript, Gross et al. have explored the idea that bactericidal (cidal) antibiotics might be more inflammatory due to lysing Gram-negative bacteria cells and releasing inflammatory materials, while bacteriostatic (static) antibiotics might be less inflammatory by containing these materials. This is explored using mouse macrophages in vitro and an in vivo peritoneal E. coli challenge model. The investigators find that while static antibiotics (carefully assessed for dose and time) make bacteria less inflammatory, cidal antibiotics (similarly carefully assessed for dose and time) make bacteria more inflammatory. This observation is most compelling in the context of the in vivo inflammatory model. Further, the investigators hypothesized that this could be due to enhanced release of LPS, but they find that this is not likely the explanation for their observations. Instead, they find that DNA release caused by cidal antibiotics and enhanced stimulation of TLR9 likely accounts for most of the effect. Overall, the study has many observations that are likely to be interesting to many investigators. However, the study as conducted has numerous shortcomings that dampen confidence.

Conceptual Comments:

In much of the study, it is hard to follow whether treatment (in vivo or in vitro) with overnight "cidal" lysed bacteria vs (for example) 3 hour "static" treated bacteria is really comparable. Is there a way to check whether both treatments have the same amount of "stuff" in them at the time of use? Maybe lyse/sonicate preparations and check if they have the same amount of LPS, DNA, etc.? I could imagine that in the different time frames if an antibiotic works even a bit slower or faster it could result in several fold changes in the end doses being used.

The investigators provide a mountain of data documenting the effectiveness of the tested antibiotics against the tested bacteria in vitro, and this is very nice. In the in vivo treatment model, do the investigators have any way of comparing the effectiveness and functions of the tested antibiotics (Tet & Cipro)? Are the viable bacteria numbers affected at similar rates? How do the interpretations change if there are large differences in bacteria number over time?

Line 87: "We hypothesized that treatment of bacteria with different types of antibiotics would impact the repertoire of PAMPs presented to macrophages and thus subsequent macrophage responses." As discussed in the introduction, this overall hypothesis is not a "new" thought, though I agree that there is still much to learn in this space. Can the overall hypothesis be formulated in a way that more specifically points to what the investigators have done in this study? Also, has the study established that the macrophage inflammatory response in vivo is the driving factor as the statement as written implies?

Presumably the macrophages kill the E. coli in the time frames being studied. How does this affect interpretation of the results? The macrophages will degrade the bacteria, so doesn't this release DNA? How would the antibiotic influence this? Is it just a timing issue (the macrophage killing takes a bit longer, maybe)? Or maybe DNA released outside activates more individual cells or different types of cells that don't necessarily ingest whole bacteria?

Fig. 1A&B relate to an in vivo treatment model, but in 1C investigators switch to an ex vivo pretreatment model. The rationale for this should be clarified in the text. Why switch to this model? Would it have been reasonable or unreasonable to measure cytokines in the treatment model where the relevant treatment effect has been observed? Does an overnight treatment with a cidal antibiotic bare any relevance to the in vivo model in which the mice are dead/dying before the bacteria are treated this long?

Technical comments:

Why switch between RAW cells and "iBMDM" in the middle of figure 2? Is there a rationale for this? Do they behave differently? I understand that the study will eventually get to the immortalized knockout cells, but why use RAW cells at all then?

In figure 3, there should be more internal controls and consistency. For example, it would be useful to see a control stimulus that is not affected by the knockouts to establish that the cells have the equivalent capacity to respond to something. One could conclude that the MyD88/TRIF knockout line was mostly dead. I presume this is not the case, but the cells responded to nothing. Further, why did the TLR2 knockout seem to affect the LPS response? Why were the Cipro bacteria affected by TLR2 knockout? Why did the TLR4 KO abrogate the reduction seen with Tet?

In figure 4, presumably the LPS response is expected to be unaffected by TLR9 knockout, so this is the control standard. Based on this it seems that the cidal treated bacteria are similar to LPS in both WT and TLR9 knockout macrophages. Conversely, it seems that the static treated bacteria are less potent than LPS in WT cells, but similarly potent in the TLR9 knockouts. This is a different conclusion than the one given. The LPS control is similarly confusing related to S6,

Please clarify statistics for figure 3A-F & 4A,B,E. What was compared to what? Averages? All possible 2-way comparisons? All data points combined? Is the comparison used statistically valid?

If the point of 3H is to test that the filtrates all have the same "level" of LPS, there should be some assurance that the assay is not topped out and is able to measure differences (e.g., dilutions showing, for example, that a 2-fold dilution reduces detected levels 2-fold).

Other Minor notes;

Line 129: "Table 1." This table is labeled "Table S1" after figures.

Line 136: "We used this MIC-adjusted approach to select antibiotic concentrations for all further experiments." Perhaps it would be good to add arrows or such to Fig. S1B to indicate the "MIC-adjusted" concentrations chosen.

I found Fig.S2 difficult to read. The data are meant to show that many antibiotics work the way the representative ones in Fig. 2 did. So, why not show the data the same way as in Fig. 2.

What are the colors in Fig. S3?

Line 172: "...determined that 6.5 hrs was optimal for balancing cytokine signal detection while minimizing any host cell death due to infection." Was there cell death due to infection? It looks minimal if anything. Is 6.5 hrs "optimal" or just a time point at which investigators have confirmed that cell death is not really a critical factor? That would be fine.

Line 177: "We saw the cidal-treated bacteria induced ~50-300% more TNF than the static-treated bacteria (Fig. 2C)." Is the minimum increase 50%? This might be worth double checking.

Line 198L "We collected 10 Gram-negative patient isolates (E. coli, Enterobacter, and Klebsiella)..." Fig. 2E. Was this 10 of each or 10 total? When we get to discussing Fig. S4 it looks like investigators mean 4 E. coli, 4 E. cloacae, and 2 K. pneumoniae?

Line 202: "...average cidal-treated bacteria (across 3 different drugs) and static-treated bacteria (across 3 different drugs)..." Which 3 drugs? The investigators have been using 4 each in the study? Which ones were omitted?

Line 256: "...levels of TNF decreased to a lower level and were similar for cidal and static treated bacterial samples, for all the antibiotic treated infections and the untreated positive control (Fig.3E)." It sure looks like the Nitro bacteria are different from the others, so it is not true to say "all the antibiotic treated". Yes, the trend is there overall.

Fig.S6C could do with a positive control for the STING deficiency (i.e., a stimulus that is STING dependent).

Line 293: "All the cidal antibiotic treated bacteria except ciprofloxacin-treated bacteria had visible amounts of DNA liberated into the soluble fraction (Fig. 4C),..." As noted later by the investigators, Cipro did not, so the word "all" should not be used.

I presume that the cytokine (and other) measurements are typically technical replicates and are usually "representative of three independent experiments." We should still know the n in each case.

Line 310: "Treatment with DNase lowered the response to cidal treated bacteria without changing the response to static treated bacteria (Fig 4E)." The statistical comparison offered is that the difference between cidal and static is lost, which is very nice. We don't, however, formally have a comparison that tells us if DNase affected the response to cidal or static treated bacteria. In fact, it looks like DNase probably had some effect on both; if so, the statement is not accurate. The text notes some toxicity of the DNase. Do the authors have any thoughts on why?

Related to figure 5, do the authors have any thoughts as to why they detect extracellular DNA by imaging Cipro treated bacteria but could not see it on gels?

Line 359: "In WT infected mice this resulted in near complete treatment failure in the ciprofloxacin (cidal) treatment group, ... (Fig 6B)." Was this a complete failure? In 6C is see the "ns" where the point is made that the cipro and tet look the same, but I don't see the relevant comparison in 1B. Really, this is a reuse of figure 1B, which is not necessary. If investigators didn't do the experiment again independently, don't show the same data again. Just refer to 1B. In any case, this in vivo TLR9 knockout experiment is pretty cool.

Line 413: "This result is consistent with recent work in the Torres lab that also found bacterial DNA to be a driver of severe infection outcomes in *S. aureus* infections." How is this consistent when this paper had nothing to do with cidal antibiotics driving inflammation? In this study, extracellular DNA is naturally a part of the infectious milieu and is an important factor in the inflammation.

Reviewer #2

(Remarks to the Author)

Within this manuscript, the authors Gross et al. present their findings regarding the evidence and consequences of bacterial DNA as a driver of TLR9-mediated inflammation during infection. This topic is relevant and interesting for publication, pending the resolution of the following critiques and questions that I have.

Line 135 – 136: While these concentrations are useful in assessing the cidal vs. static actions of the antibiotics, how do they relate to physiological levels that are typically achieved at the sites of infection? Static drugs at higher concentrations may demonstrate killing at sufficient levels.

Line 144-145: How are PK/PD differences being accounted for in this experiment? Is the cidal drug being retained for a similar amount of time as the static drug in this experiment?

Fig 1B: It would be useful to include data of the bacterial burdens in the different mouse niches, or at least from the peritoneal cavity, to demonstrate differences between the cidal and static drug and show whether decreased survival is associated with higher bacterial titers. This would be especially interesting if this is not the case.

Fig 1C: Why only look up to 2hr post-antibiotic treatment? This seems to not account for different drug elimination half-lives, onset of action, and tissue distribution of the two antibiotics. This is another reason why bacterial burden might be important to look at in this figure. Could it be that mice are dying within a few hours due to one of these other pharmacokinetic parameters, rather than just cidal vs. static mechanisms?

Line 160 – 178: How much does the drug class matter in influencing TNF production? If ciprofloxacin, for example, can potentiate TNF release from macrophages in response to LPS (doi: 10.1111/j.1365-2249.1991.tb05728.x.), then we would also need to separate the TNF contributions of bacterial lysis due to antibiotic action vs. antibiotic modulation of TNF release. Did you treat/pre-treat host cells with antibiotics before bacterial infection?

Figure 2D: The x axis labels are not aligned. Also, It would be nice to show this same figure except using dose-dependent killing with ciprofloxacin as a way to show that TNF concentration is proportional to amount of bacterial killing, regardless of mechanism. If this is not the case, it would reveal that other factors may be in play.

Line 252 – 262: If more DNA is liberated into the soluble fraction with most of the cidal drugs, I think there should be some discussion on why we don't see an increase in TNF levels in macrophages treated with the cidal supernatant. One would think that there should be more "inflammatory soluble factors" in the form of the liberated DNA with the LPS levels being roughly equivalent.

Line 336: Why do we see increased extracellular DNA for cipro in this figure compared to the supernatant in the previous figure 4C? Is it just a difference in the timepoint?

Line 417: antibiotics is misspelled.

Line 441 - 447: Are there any clinical meta-analyses comparing human outcomes in infection-mediated systemic inflammatory response syndrome or the like addressed with a static broad spectrum antibiotic(s) over a cidal one?

It is important to recognize in this experiment that the mice were being infected suddenly with a large bacterial inoculum and just as suddenly treated with a large antibiotic dose. I do wonder how that might differ from mice that get sick more gradually that have been enduring a growing infection for hours or days. This could be worth mentioning or otherwise addressing in the discussion.

Reviewer #3

(Remarks to the Author)

NCOMMS-24-24597

Here, Gross et al undertake a careful and elegant study unpicking the role of released bacterial DNA in damaging inflammation following bactericidal drug treatment. Using an in vivo mouse model of acute peritonitis, the authors noted a disparity in the efficacy of treatment with cidal versus static drugs – with static drugs considerably more protective than cidal. Further studies confirmed that bacteria treated with cidal drugs induce higher levels of cytokines, particularly TNF (as a marker for inflammation), from infected macrophages. This inflammatory response is associated with bacterial DNA released following cidal treatment and is dependent upon the TLR-9 signalling pathway. This TLR-9 mediated inflammatory response is correlated with the failure of cidal drugs to treat the acute model of infection supported by the fact that efficacy of these drugs improves dramatically in TLR-9 null mice. These studies are generally well carried out and have broad implications for the future selection of drug treatments for high level bacterial infections. The authors themselves note that their observations are from a single in vivo model of infection – so they may want to be slightly more careful and tone down the extrapolation of this hypothesis more broadly. In addition, I have some further points that they should address.

1. Can the authors please expand on the specifics/mechanism of how they think elevated TNF results in cidal drug failure?
2. As a non-immunologist – can you explain if TLR9 is known to “sense” anything outside of DNA? This may help non-specialist readers.

3. Treatment with static drugs to prevent a damaging immune response following destruction of a pathogen is not a novel concept. For instance, drugs that target eggs in schistosomiasis infections are contraindicated due to the damaging immune response that may result from a release of antigens. The authors may want to include some examples of such phenomenon in their discussion.

4. Why did the authors switch between cEC1 and K12 in the experiments shown in Figs 1 and 2?

5. Fig 4 – did the authors attempt to quantify the bacterial DNA released? Would be interesting to know a ballpark/bacteria. One could argue that such a number would be useful in deciding when and when not to apply cidal/static drugs for treatment.

Minor

In Table 1 (or S1) – called both in the text – can the authors add which drugs are cidal versus static and also their precise mechanism of action (ie inhibits protein synthesis).

Version 1:

Reviewer comments:

Reviewer #1

(Remarks to the Author)

In general, the responses to my queries are sufficient.

Reviewer #2

(Remarks to the Author)

All my previous concerns were addressed appropriately.

NCOMMS-24-24597

Bactericidal Antibiotic Treatment Induces Damaging Inflammation via TLR9 Sensing of Bacterial DNA. Gross et al.

Response to reviewers. Author responses are in blue bold underlined text. All line numbers refer to the clean text version (line numbers differ in the tracked text).

We thank the reviewers for their comments. We feel they have allowed us to significantly improve the quality of the manuscript in several key ways. We have added new data in panels C and E to main Figure 1, panels C and E to main Figure 6, an entirely new supplementary Figure S6 A-F, and panel A to supplementary Figure S7: 11 new data panels in total. We have also made reviewer-suggested changes that we hope will help improve clarity in Table 1 and supplementary Figures S1-S3, and added important points to the discussion that we believe situate this study better in the broader field.

Comments from Reviewer 1:

In this manuscript, Gross et al. have explored the idea that bactericidal (cidal) antibiotics might be more inflammatory due to lysing Gram-negative bacteria cells and releasing inflammatory materials, while bacteriostatic (static) antibiotics might be less inflammatory by containing these materials. This is explored using mouse macrophages in vitro and an in vivo peritoneal E. coli challenge model. The investigators find that while static antibiotics (carefully assessed for dose and time) make bacteria less inflammatory, cidal antibiotics (similarly carefully assessed for dose and time) make bacteria more inflammatory. This observation is most compelling in the context of the in vivo inflammatory model. Further, the investigators hypothesized that this could be due to enhanced release of LPS, but they find that this is not likely the explanation for their observations. Instead, they find that DNA release caused by cidal antibiotics and enhanced stimulation of TLR9 likely accounts for most of the effect. Overall, the study has many observations that are likely to be interesting to many investigators. However, the study as conducted has numerous shortcomings that dampen confidence.

We thank the reviewer for their thoughtful consideration, and for noting that our data are likely to be interesting to many investigators. We hope the new data we have added can thoroughly address the shortcomings identified by this reviewer.

Conceptual Comments:

1. In much of the study, it is hard to follow whether treatment (in vivo or in vitro) with overnight “cidal” lysed bacteria vs (for example) 3 hour “static” treated bacteria is really comparable. Is there a way to check whether both treatments have the same amount of “stuff” in them at the time of use? Maybe lyse/sonicate preparations and check if they have the same amount of LPS, DNA, etc.? I could imagine that in the different time frames if an antibiotic works even a bit slower or faster it could result in several fold changes in the end doses being used.

This is an important point, and we apologize that we did not make this clearer in the original submission. The only experiments in which we varied the timing of antibiotics between cidal and static conditions were the *in vivo* cytokine quantification experiments in the original Figs 1C, and 6D (current Figs 1F and 6F). We designed this experiment this way to ensure that the mice receiving cidal treated bacteria received 100% dead bacteria, and the mice receiving static treated bacteria received 100% growth-halted bacteria (see new Fig 1E/ Fig 6E for bacterial CFU quantification supporting this). When we attempted shorter treatment intervals, we had incomplete killing with the cidal drug. When we attempted longer treatment intervals, we saw either growth or some killing with the static drug. We therefore used different timing intervals in this model to ensure that both types of drugs produced their characteristic effects: killing for cidal, and growth halting for static. These drugs working as intended likely will alter the amount of PAMPs available to be sensed by the host, as we believe differential exposure to bacterial DNA to be the mechanistic basis underlying the differential inflammatory responses induced by the different drug treated bacteria. In the other *in vivo* survival studies in these figures (Figs 1A-B and 6A-C in the original draft, now Figs 1A-B and Figs 6A-B) all bacteria were administered in identical quantities from the same tube, and cidal and static drugs were then administered in a single dose at a fixed time 30 min post infection.

We have made the following changes to clarify these points in the paper: 1. Updated Figures 1 and 6 to include two different schematics to better differentiate the different models (original 1C, and 6E respectively, now new Fig 1D, and new Fig 6D) that we've called the 'pathogenesis' and 'acute inflammatory response' models respectively. 2. Added new Fig 1E/6E to display the CFU plating we did to quantify inocula going into the mice in the *ex vivo* acute inflammatory response model. These counts validate that using the chosen time intervals allowed the drugs to work as intended: 100% killing for the cidal, and 100% growth stoppage for the static. 3. Updated the results text to better clarify that the pathogenesis infection model is intended to elucidate whether there are survival differences that arise downstream of the different treatments (see line 140-141, pathogenesis infection model section below), whereas the acute inflammatory response *ex vivo* treatment model is intended to assess the dynamics of acute host cytokine responses elicited by identical doses of bacteria either pre-killed or pre-growth arrested with these drugs (see line 156-169, acute inflammatory response model section below).

Pathogenesis infection model updated description: "To determine how different types of antibiotic treatments impact infection pathogenesis outcomes, we developed an *in vivo* peritonitis infection model comparing cidal-treated (cipro), static-treated (tet), and untreated mice."

Acute inflammatory response infection model updated description: "To assess if differential host inflammatory responses could be an important mechanism driving these survival differences, we next measured representative proinflammatory

serum cytokine levels in WT mice one and two hrs after they received equivalent amounts of either cidal antibiotic-treated cEC1 E. coli bacteria or static treated cEC1 E. coli bacteria via intraperitoneal (IP) injection (Fig. 1D). To assess the efficacy of these antibiotic treatments, we quantified the inocula prior to injection: this quantification verified that the treated bacteria were either completely killed (cidal) or completely growth arrested (static) (Fig. 1E). ...These results are consistent with the survival data (Fig. 1B), and show that different types of antibiotic treatments that ultimately have roughly equivalent effects on bacterial clearance (Fig. 1C) can cause different degrees of detrimental inflammation in vivo.”

New acute inflammatory response model schematics (1D left/6D right):

Legend: (D) Schematic representation of ex vivo cytokine quantification experiment. cEC1 bacteria were treated until either completely killed (cidal drugs) or growth halted (static drugs), then used in identical quantities to infect mice. Cytokines were quantified 1 and 2 hours later.

New CFU quantification of the inputs in this model (1E left, 6E right):

Legend: Plating of ex vivo inocula prior to beginning infections. Six samples were collected and dilution plated from each culture; data shown is representative of three independent experiments.

2. The investigators provide a mountain of data documenting the effectiveness of the tested antibiotics against the tested bacteria in vitro, and this is very nice. In the in vivo treatment model, do the investigators have any way of comparing the effectiveness and

functions of the tested antibiotics (Tet & Cipro)? Are the viable bacteria numbers affected at similar rates? How do the interpretations change if there are large differences in bacteria number over time?

We appreciate the reviewer's positive characterization of our *in vitro* antibiotic validation data. To address the concern about *in vivo* antibiotic kinetics/efficacy, we quantified bacterial load in a peritoneal lavage, the spleen, and the left liver lobe in these mice under several conditions: 5 min post infection (but before antibiotic treatment), and then untreated, cipro treated, and tet treated at 4 hrs post infection. We observed that both drugs were largely, and similarly, effective in controlling bacterial load over time. We have updated Figure 1 and Figure 6 to include these counts for WT and TLR9 KO mice respectively (see new Fig 1C and 6C) and highlighted these data and our interpretations in the text (see lines 147-154 and below).

New Figure 1C (WT mice):

New Figure 6C (TLR9 KO mice):

Legend: (C) Bacterial burdens were quantified by plating from CFU a wash of the peritoneal lavage (PL), and homogenates of the spleen and the left lobe of the liver at 5 min post infection (before antibiotic treatments), and after 4 hours after the indicated treatments. Data are pooled from groups of 6 mice (3 male, 3 female) across two independent experiments and shown in full.

This section now includes: “Additionally, we measured bacterial burdens in a peritoneal lavage, the left lobe of the liver, and the spleen four hours post infection in this model (Fig. 1C). Both drugs reduced bacterial burden (by ~2-4 log-fold) in all

three anatomical sites compared to pre-treatment input levels, and bacterial burdens in untreated control mice; leading to similar burdens in the later time points in both antibiotic treated groups. Together these results highlight the importance of antibiotic selection in influencing the survival outcome of these infections, and suggest that there are mechanisms beyond differential bacterial control accounting for the large survival difference we observe between cidal and static treated groups.”

3. Line 87: “We hypothesized that treatment of bacteria with different types of antibiotics would impact the repertoire of PAMPs presented to macrophages and thus subsequent macrophage responses.” As discussed in the introduction, this overall hypothesis is not a “new” thought, though I agree that there is still much to learn in this space. Can the overall hypothesis be formulated in a way that more specifically points to what the investigators have done in this study? Also, has the study established that the macrophage inflammatory response *in vivo* is the driving factor as the statement as written implies?

We have adapted this introductory sentence to more specifically indicate the mechanistic types of antibiotics being compared, and to allow for more breadth of immune responses as possibly driving our *in vivo* phenotypic differences (see text changes lines 87-90 and below).

This sentence now reads: “We hypothesized that treatment of bacteria with different types of antibiotics (namely cidal antibiotics that kill the bacteria vs. static antibiotics that do not) would impact the repertoire of PAMPs presented to immune cells, and thus subsequent immune responses.

4. Presumably the macrophages kill the *E. coli* in the time frames being studied. How does this affect interpretation of the results? The macrophages will degrade the bacteria, so doesn't this release DNA? How would the antibiotic influence this? Is it just a timing issue (the macrophage killing takes a bit longer, maybe)? Or maybe DNA released outside activates more individual cells or different types of cells that don't necessarily ingest whole bacteria?

This is an interesting point. If we understand correctly, the reviewer is wondering if bacterial DNA would be released by both cidal drug mediated bacterial destruction and by macrophage mediated killing of static treated bacteria. We noted in preliminary work that bacterial lysis by cidal antibiotics often requires bacteria to be actively replicating and to have that replication perturbed. Growth halted bacteria are often resistant to the effects of antimicrobials that lead to lysis (this is why static drugs are often known to inhibit the activity of cidal drugs when co-administered).^{1,2} Thus, it is possible that static treated, i.e. growth halted, bacteria are actually more resistant to macrophage-induced lysis than untreated bacteria, which would mean static treated bacteria may not release DNA after engulfment by macrophages, which would be consistent with the lower inflammatory response observed with static-treated bacteria. We also agree with the reviewer that externally released DNA resulting from cidal lysis of bacteria would likely have the potential to activate more cells and cell types.

5. Fig. 1A&B relate to an in vivo treatment model, but in 1C investigators switch to an ex vivo pretreatment model. The rationale for this should be clarified in the text. Why switch to this model? Would it have been reasonable or unreasonable to measure cytokines in the treatment model where the relevant treatment effect has been observed? Does an overnight treatment with a cidal antibiotic bare any relevance to the in vivo model in which the mice are dead/dying before the bacteria are treated this long?

We have addressed this comment in part in our responses to points 1 and 2 above. Using our infection pathogenesis model (Fig 1A-B) we determined that a cidal antibiotic and a static antibiotic treatment drive different survival outcomes, despite achieving similar levels of bacterial clearance (see new Fig 1C and page 4). To better clarify the rationale for the switch to the second, acute inflammatory response ex vivo treatment model (now with its own schematic, see new Fig 1D and page 3): our goal was to try to tease apart what factors could explain the differences observed in survival. We wondered if the antibiotic treatment had an impact on cytokines, and, to get as clear an answer as possible, we wanted to ensure that all the cidal treated bacteria were dead, and all the static treated bacteria were growth halted (see new Fig 1E and page 3). The only way to ensure this was to treat with drug first, before bacteria went into the mouse. The relevance of these data are that these experiments revealed dramatic *in vivo* differences in host cytokine responses to equal amounts of bacteria treated with the different antibiotics. This led us to hypothesize that differential host responses could play an important mechanistic role in determining the 30-hour survival differences we observed in the pathogenesis model, and informed the direction of further efforts to investigate that difference in the rest of the paper.

Technical comments:

1. Why switch between RAW cells and “iBMDM” in the middle of figure 2? Is there a rationale for this? Do they behave differently? I understand that the study will eventually get to the immortalized knockout cells, but why use RAW cells at all then?

The initial screening experiment in Figure 2 (using K12 and RAW cells) was the first experiment we did in this project. As the project developed, we transitioned away from both laboratory bacterial strains and RAW cells towards more translationally relevant clinically-derived bacterial strains and primary macrophages (as well as immortalized primary macrophages to ask specific mechanistic questions). If the reviewer feels the initial screen data detracts from the paper as a result of switching the cells, we can move it to the supplement or remove it.

2. In figure 3, there should be more internal controls and consistency. For example, it would be useful to see a control stimulus that is not affected by the knockouts to establish that the cells have the equivalent capacity to respond to something. One could conclude that the MyD88/TRIF knockout line was mostly dead. I presume this is not the case, but the cells responded to nothing. Further, why did the TLR2 knockout seem to affect the LPS response? Why were the Cipro bacteria affected by TLR2 knockout? Why did the TLR4 KO abrogate the reduction seen with Tet?

To address the concern about the MyD88/TRIF KO cells, we have added an interferon stimulation and assessed response from the stimulated cells by measuring pSTAT1 levels via Western blot (see new Fig S6B, and new text lines 266-270 and below). As is shown in that figure, both WT and MyD88/TRIF KO cells can respond to interferon beta by up regulating pSTAT1 to a similar degree.

We have added this text to results: “To determine whether MyD88/Ticam-1 ^{-/-} iBMDM cells are capable of responding to (non-TLR) stimuli, we assessed pSTAT1 protein expression in response to a 30-minute interferon beta stimulation. Both WT and MyD88/Ticam-1 ^{-/-} iBMDM cells had comparable responses to interferon stimulation (Fig. S6B).”

New Fig S6B:

Legend: WT and STING KO iBMDMs were stimulated with mouse interferon beta for 30 min. pSTAT1 protein expression was quantified in each cell type by Western blot both with and without interferon.

We see some reduction in KLA signaling relative to WT KLA signaling in several TLR KO cells (including TLR2). We do not know for certain why this occurs, but we would speculate that it could be due to the fact that these receptors share the majority of downstream signaling components, and that KO of one receptor can affect the response through another. In fact, there is a considerable literature on signaling crosstalk between TLR pathways that includes both suppression and synergy among co-activated receptors.³⁻⁵ It is also worth noting that a recent impressive paper from the Kagan laboratory⁶ demonstrated that the proximal Myd88 adapter (that is shared by most TLRs including TLR4, 2 and 9) forms a large so-called ‘myddosome’ that contains the multiple branches of the downstream TLR signaling pathway (NF-kB, MAPK and IRFs) in a single multiprotein complex. It is therefore conceivable that perturbation of one TLR could influence responses through another.

It is not entirely clear why macrophages infected specifically with cipro-treated bacteria lose a large portion of their TNF response compared to WT in the absence of TLR2, though it is possible that these specific bacteria signal more through TLR2 in addition to other TLRs in WT cells. We have updated the text to note this possibility (see lines 274-278 and below). That being said, observing cipro equalize with the other drugs in the absence of TLR9 (Fig 4B) or presence of DNase (Fig 4E), and seeing the *in vivo* survival difference between cipro and tet treated mice

drop from 67% in WT mice to 10.3% in TLR9 KO mice (Fig 6B), gives us confidence that signaling differences through TLR9 are the primary driver of the cidal vs. static phenotype we observe.

New text reads: “There is however some variability in this trend with respect to individual drugs – notably, in the absence of TLR2 macrophages lost some response to specifically cipro treated infections (Fig. 3C), possibly indicating that cipro treated bacteria natively signal through multiple TLR pathways, and may release more bacterial lipoproteins (BLPs) than other cidal treatments.”

To investigate the pattern the reviewer notes in the TLR4 KO infections, we have repeated the TLR4 KO cell comparison several additional times. We see that tetracycline treated bacteria are generally not able to induce much TNF response in this system (they induce similar TNF responses to the other 3 static drug treated bacteria). We have replaced Fig 3C with a more representative experiment.

New Fig 3C:

Legend: (A-D) TNF quantified by ELISA at 6.5hr from WT (A), *MyD88/Ticam-1/-* (*MyD88/TRIF*) (B), *Tlr4/-* (C), and *Tlr2/-* (D) iBMDM macrophages infected with media alone (black), bacteria without antibiotics (black), KLA (orange), and equivalent concentrations of cidal (blue) and static treated bacteria (red) as indicated.

3. In figure 4, presumably the LPS response is expected to be unaffected by TLR9 knockout, so this is the control standard. Based on this it seems that the cidal treated bacteria are similar to LPS in both WT and TLR9 knockout macrophages. Conversely, it seems that the static treated bacteria are less potent than LPS in WT cells, but similarly potent in the TLR9 knockouts. This is a different conclusion than the one given. The LPS control is similarly confusing related to S6.

Though we may expect LPS response to be entirely consistent across cell lines, in fact we observe some reduction in signaling downstream of LPS in TLR KO cell lines across the board (including TLR9), and we have elaborated on possible reasons for this in the response to point 2 above. To ascertain the true extent of

this effect for the TLR9 KO cells specifically, we repeated this experiment several more times, which revealed that the degree of loss of signaling from KLA in the TLR9 KO cells appeared to be unusually high in the experiment presented in the original submission. To correct for this, we replaced Fig 4B with data showing an aggregated response across multiple experimental replicates (below).

New Fig 4B:

Legend: TNF quantified by ELISA at 6.5 hr from WT (A) or Tlr9^{-/-} (B) iBMDM macrophages infected with media alone (black), bacteria without antibiotics (black), KLA (orange), and equivalent concentrations of cidal (blue) and static treated bacteria (red) as indicated.

We also appreciate this reviewer’s thoughtful refinement of our conclusions with respect to the movements of the individual drug groups in the TLR9 KO model in relation to the WT TNF responses, and we have updated the text to reflect them (see lines 308-313, and below).

Revised text reads: “We found that when we infect iBMDMs lacking TLR9 with cidal treated bacteria we observe diminished TNF signaling relative to the corresponding infections of WT iBMDMs, whereas signaling from static-treated bacteria was maintained (if anything we observe a slight increase in TNF signal in the TLR9 ^{-/-} iBMDMs relative to WT iBMDMs). The cidal vs. static difference we observe in WT infections, however, was dramatically reduced in Tlr9 ^{-/-} BMDMs (Fig. 4A-B).”

4. Please clarify statistics for figure 3A-F & 4A,B,E. What was compared to what? Averages? All possible 2-way comparisons? All data points combined? Is the comparison used statistically valid?

In all these cases we have compared the data from all cidal treated bacteria as a group, to all static treated bacteria as a group. If we understand the reviewer correctly, we believe this approach would equate to “averages” and “all data points combined”, as we used the underlying individual datapoints from each drug in each panel to form the cidal and static groupings that we reference when discussing results for Figs 2C, 3A-F, and 4A, B and E. To address this comment,

we have clarified the relevant figure legends to more explicitly state these comparisons, as well as adding additional clarifying text in lines 259-261 (see below).

New text reads: “Mirroring the analysis in Fig 2C, we compared the effects of cidal treatments as a group (combined from infecting with each cidal drug) to the effects of static treatments as a group (combined from infecting with each static drug).”

5. If the point of 3H is to test that the filtrates all have the same “level” of LPS, there should be some assurance that the assay is not topped out and is able to measure differences (e.g., dilutions showing, for example, that a 2-fold dilution reduces detected levels 2-fold). We have addressed this comment by adding a dilution series to the newly created Sup Fig 6, see panels C-F and text lines 296-299 (see below). These data confirm that the consistent level of LPS detected downstream of the different treatments (and of no treatment) is not a saturation artifact in the original assay. Due to day by day variations in the absolute (though not relative) amounts of detector observed, and the lack of a manufacturer provided standard curve for this assay, we have normalized both these plots and the original data in Fig 3G-H to the negative control values from each run.

New Fig S6 C-F:

Legend: “(C-F) WT iBMDM macrophages were infected with media alone (black), bacteria without antibiotics (black), KLA (orange), and equivalent concentrations of cidal (blue) and static treated bacteria (red) as indicated. Resulting supernatants were collected and applied to TLR4 reporter HEK cells: undiluted (C), 3-fold diluted (D), 9-fold diluted (E), and 27-fold diluted (F), and Quanti-blue detector reagent is quantified.”

New text says: “To determine whether these equal levels were a result of assay saturation, we also diluted these supernatants and quantified the Quanti-blue reporter response to the diluted supernatants (Fig. S6 C-F). We observed corresponding reductions of the responses (Fig. S6 C-F).”

Other Minor notes:

1. Line 129: “Table 1.” This table is labeled “Table S1” after figures. We have corrected this to consistently refer to it as Table 1.

2. Line 136: “We used this MIC-adjusted approach to select antibiotic concentrations for all further experiments.” Perhaps it would be good to add arrows or such to Fig. S1B to indicate the “MIC-adjusted” concentrations chosen.

We have added arrows at the bottom of the Fig S1B plots to indicate the antibiotic concentrations used.

3. I found Fig.S2 difficult to read. The data are meant to show that many antibiotics work the way the representative ones in Fig. 2 did. So, why not show the data the same way as in Fig. 2. What are the colors in Fig. S3?

In Fig S2 we show TNF readouts in each case as fold change, normalized to the relevant MOI-matched, infected, untreated, contemporaneously processed control. To make this clearer, we have now labeled the Fig S2 scale bar: Fold Change (normalized to untreated). With this normalization, all conditions less than 1 indicate suppression relative to control, and all conditions greater than 1 indicate enhancement relative to control. We use the color gradient to indicate trends in the magnitude of the fold change of suppression/enhancement of TNF responses that we see across these infections.

The colors in S3 are cells infected with different bacterial strains. We have updated the legend to include this information.

4. Line 172: “...determined that 6.5 hrs was optimal for balancing cytokine signal detection while minimizing any host cell death due to infection.” Was there cell death due to infection? It looks minimal if anything. Is 6.5 hrs “optimal” or just a time point at which investigators have confirmed that cell death is not really a critical factor? That would be fine.

We did observe more substantive cell death when extending this time course out to 12 hours with certain clinical strains (though not cEC1), and thus chose to conduct all our comparative experiments at 6.5 hours when – as the reviewer correctly notes – macrophage cell death due to infection was minimal to non-existent.

5. Line 177: “We saw the cidal-treated bacteria induced ~50-300% more TNF than the static-treated bacteria (Fig. 2C).” Is the minimum increase 50%? This might be worth double checking.

We thank the reviewer for bringing this to our attention, as those figures were from a predecessor to the final graph shown in 2C. In this experiment, using the average across replicates for the amount of TNF from each drug, we see an average ~2.18 fold (118% increase) in TNF from cidal treatments as compared to static treatments (all the cidals averaged / all the statics averaged), with a range of 3.71 fold (271% increase) (meropenem, highest cidal / doxycycline, lowest static) to 1.23 fold (23% increase) (ciprofloxacin, lowest cidal / chloramphenicol, highest static). We have updated the results text to include appropriate numbers from this specific replicate, and the Figure 2 legend to include better descriptions to clarify what is being compared to what when we report out both the 118% average difference

between the groups, and the range of differences observed with the most extreme point examples (see lines 190-191 below, and new lines 771-776 in the Fig 2 legend).

Updated results text now says: “We saw the cidal-treated bacteria induced ~23-271% more TNF than the static-treated bacteria, with a 118% increase on average. (Fig. 2C).”

Updated Figure 2 legend text: “...statistical comparisons in 2C are of cidal drugs (as a group) to static drugs (as a group). We compute the average in percent increase as: (the average of all the cidal / the average of all the statics) – 1. To identify the reported range of percent increases, we computed the minimum difference (lowest cidal, ciprofloxacin, / highest static, chloramphenicol) – 1, and the maximum difference (highest cidal, meropenem / lowest static, doxycycline) – 1.”

6. Line 198L “We collected 10 Gram-negative patient isolates (E. coli, Enterobacter, and Klebsiella)...” Fig. 2E. Was this 10 of each or 10 total? When we get to discussing Fig. S4 it looks like investigators mean 4 E. coli, 4 E. cloacae, and 2 K. pneumoniae?
10 total. We have updated the text to list the # of strains considered with each specific species we mention (see lines 211-212, and below).

New text reads: “(E coli (n = 4), Enterobacter (n = 4) and Klebsiella (n = 2).”

7. Line 202: “...average cidal-treated bacteria (across 3 different drugs) and static-treated bacteria (across 3 different drugs)...” Which 3 drugs? The investigators have been using 4 each in the study? Which ones were omitted?
We updated the parentheses (lines 216-218) to list the specific drugs of each type we tested in this experiment. The reason ceftriaxone and nitrofurantoin were not assessed in these panels is that this was a very early experiment (the first with a range of clinical strains), and at that point we had not yet begun adding ceftriaxone and nitrofurantoin to our standard panel of drugs to assess.

8. Line 256: “...levels of TNF decreased to a lower level and were similar for cidal and static treated bacterial samples, for all the antibiotic treated infections and the untreated positive control (Fig.3E).” It sure looks like the Nitro bacteria are different from the others, so it is not true to say “all the antibiotic treated”. Yes, the trend is there overall.
Thank you for pointing this out. We have clarified the text to note the overall trend, and explicitly note the drop from nitrofurantoin-treated bacteria (see lines 285-287, and below).

Updated text: “This filter experiment suggests a trend of similar levels of independently inflammatory soluble factors across the different cidal and static treatments, with a decrease in those levels for nitro-treated bacteria.”

9. Fig.S6C could do with a positive control for the STING deficiency (i.e., a stimulus that is STING dependent).

We concur with the reviewer, and have updated Figure S7 to include protein expression data for WT (STING expressing) and STING KO (STING non-expressing) cells that clearly demonstrates the absence of STING expression in our STING KO cell line; see newly added panel S7A. We have also updated the text and figure legends with the newly added panel (see lines 908-909 and below), and changed all the corresponding lettering in Fig S7.

New Fig S7A:

Updated legend text: “(A) STING protein expression quantified by Western blot in WT and STING KO iBMDMs.”

10. Line 293: “All the cidal antibiotic treated bacteria except ciprofloxacin-treated bacteria had visible amounts of DNA liberated into the soluble fraction (Fig. 4C),...” As noted later by the investigators, Cipro did not, so the word “all” should not be used.

We have changed the wording to nearly all (lines 326-328 and below), reflecting the fact that this was the case for all but one cidal drug.

Updated text: “Nearly all the cidal antibiotic-treated bacteria (with the exception of ciprofloxacin-treated bacteria) had visible amounts of DNA liberated into the soluble fraction (Fig. 4C)...”

11. I presume that the cytokine (and other) measurements are typically technical replicates and are usually “representative of three independent experiments.” We should still know the n in each case.

This is correct. Figure legends have been updated to note both the n within each experiment, and the # of independent experiments.

12. Line 310: “Treatment with DNase lowered the response to cidal treated bacteria without changing the response to static treated bacteria (Fig 4E).” The statistical comparison offered is that the difference between cidal and static is lost, which is very nice. We don’t, however, formally have a comparison that tells us if DNase affected the response to cidal or static treated bacteria. In fact, it looks like DNase probably had some effect on both; if so, the statement is not accurate. The text notes some toxicity of the DNase. Do the authors have any thoughts on why?

We concur with the reviewer and have changed this sentence to "Treatment with DNase abrogated the difference between cidal and static treatments that we observed in untreated cells (Fig. 4E)." , see updated lines 344-345.

Although we are not certain of the mechanism, we are not the only group to note some degree of cellular toxicity of this family of DNAses when combined with TNF signaling (<https://pubs.acs.org/doi/10.1021/bi001041a>)⁷, though in this paper they express it endogenously whereas we are applying the enzyme exogenously under conditions expected to produce endogenous TNF signaling.

13. Related to figure 5, do the authors have any thoughts as to why they detect extracellular DNA by imaging Cipro treated bacteria but could not see it on gels?

We assessed bacteria treated for 3 hrs on the gel vs. bacteria treated for 6 hrs in the imaging assays. When we attempted to treat for a longer time prior to the gel assay we ran into issues with DNA sample purity from the extracts that we did not have with the more self-contained imaging assays, where we stained in the infection environment.

14. Line 359: "In WT infected mice this resulted in near complete treatment failure in the ciprofloxacin (cidal) treatment group, ... (Fig 6B)." Was this a complete failure? In 6C is see the "ns" where the point is made that the cipro and tet look the same, but I don't see the relevant comparison in 1B. Really, this is a reuse of figure 1B, which is not necessary. If investigators didn't do the experiment again independently, don't show the same data again. Just refer to 1B. In any case, this in vivo TLR9 knockout experiment is pretty cool.

We have removed the redundant graph from Fig 6, and are now showing only the TLR9 KO mice in Figure 6. In terms of the original data in Fig 1B, we make the comparison between cipro treatment and tet treatment with a vertical line at the far right, with the p-value indicated on top. We do believe that reduced treatment efficacy from ~75% to ~11% constitutes near complete treatment failure (particularly because the no-treatment survival rate was ~5%, so the baseline is not 0%). We are glad this reviewer shares our enthusiasm for this particular experiment!

15. Line 413: "This result is consistent with recent work in the Torres lab that also found bacterial DNA to be a driver of severe infection outcomes in S. aureus infections." How is this consistent when this paper had nothing to do with cidal antibiotics driving inflammation? In this study, extracellular DNA is naturally a part of the infectious milieu and is an important factor in the inflammation.

We feel that our observations are consistent in the sense that they reinforce the importance of bacterially derived extracellular DNA to a hosts' inflammatory response to the infection. In the cited case, the said eDNA was naturally occurring, and in our case, it comes about as a result of using specifically cidal treatments against a different strain of bacteria. In both cases though, more bacterial extracellular DNA leads to increased inflammatory output from the host immune system.

Comments from Reviewer 2:

Within this manuscript, the authors Gross et al. present their findings regarding the evidence and consequences of bacterial DNA as a driver of TLR9-mediated inflammation during infection. This topic is relevant and interesting for publication, pending the resolution of the following critiques and questions that I have.

We appreciate the reviewer's assessment that this is a relevant and interesting topic for publication, and hope the additional data and clarifications we added have fully addressed outstanding questions.

1. Line 135 – 136: While these concentrations are useful in assessing the cidal vs. static actions of the antibiotics, how do they relate to physiological levels that are typically achieved at the sites of infection? Static drugs at higher concentrations may demonstrate killing at sufficient levels.

We selected these concentrations for *in vivo* dosing regimens based on previous literature comparison, and guidance from our veterinary staff and IACUC committee in determining what levels of these drugs are "typically" effective in treating mice with *E. coli* infections. This analysis did not include quantifying exact drug levels achieved at various sites of systemic infection over time, so we do not know precisely how they compare to physiological levels in patients. However, based on the CFU counts included in this revision (see new Fig 1C, new Fig 6C, and reproduced above on page 4), both treatments at these doses appear to be roughly equally effective at restraining otherwise unchecked bacterial growth *in vivo* in the peritoneal cavity, liver, and spleen.

2. Line 144-145: How are PK/PD differences being accounted for in this experiment? Is the cidal drug being retained for a similar amount of time as the static drug in this experiment?

This is a good question. We are not accounting for PK/PD differences in this experiment specifically. However, as noted above in response to the previous comment, these antibiotic treatments do yield similar bacterial burdens at 4 hrs in a peritoneal wash, liver, and spleen (new Fig 1C/6C).

3. Fig 1B: It would be useful to include data of the bacterial burdens in the different mouse niches, or at least from the peritoneal cavity, to demonstrate differences between the cidal and static drug and show whether decreased survival is associated with higher bacterial titers. This would be especially interesting if this is not the case.

We have addressed this question by adding bacterial load data from the peritoneal cavity, left liver lobe, and spleen in new Fig 1C and new Fig 6C (see response to reviewer 1's second comment for more extended discussion on this point, page 4-5).

4. Fig 1C: Why only look up to 2hr post-antibiotic treatment? This seems to not account for different drug elimination half-lives, onset of action, and tissue distribution of the two antibiotics. This is another reason why bacterial burden might be important to look at in this figure. Could it be that mice are dying within a few hours due to one of these other pharmacokinetic parameters, rather than just cidal vs. static mechanisms?

We opted to quantify cytokines at 2 hrs because during initial optimization experiments we observed peak host TNF response at 90 min - 2 hrs. The mice are not dying at 2 hrs, we euthanize them to collect enough blood to quantify cytokines. Additionally, in this model (originally 1C, now 1F and titled the acute inflammatory response model) we administer drugs to the bacteria *ex vivo* (and verify that they work as intended: complete killing for cidals, complete growth inhibition without killing for statics, see new Fig 1E, page 3) before infecting mice with the pre-treated bacteria.

5. Line 160 – 178: How much does the drug class matter in influencing TNF production? If ciprofloxacin, for example, can potentiate TNF release from macrophages in response to LPS (doi: 10.1111/j.1365-2249.1991.tb05728.x.), then we would also need to separate the TNF contributions of bacterial lysis due to antibiotic action vs. antibiotic modulation of TNF release. Did you treat/pre-treat host cells with antibiotics before bacterial infection? To address this concern, we assessed TNF output after incubating WT iBMDMs with each of the drugs +/- KLA. We did not observe increases in TNF release when we treated iBMDMs with any of the eight antibiotics in our panel – either alone, or in combination with KLA stimulation for 6.5 hrs (see new Sup Fig 6A, and new text lines 255-259 and below). In the cell comparison experiments in Fig 2-4, we did not pre-treat the host cells with antibiotics before infecting them with antibiotic treated bacteria, so the longest time interval host cells received antibiotics for in any of these experiments is 6.5 hours.

New Fig S6 A:

A

Legend: (A) TNF quantified by ELISA at 6.5 hr from iBMDMs that were incubated with each individual antibiotic at the concentrations we used for our cell comparisons in the absence of bacteria with no KLA (left graph), and with KLA (right graph). One representative experiment is shown of 3 independent experiments (3 technical replicates were run per condition per experiment).

New text reads: “To determine if the antibiotics alone (in the absence of bacteria) had any effect on iBMDM TNF production, we added each antibiotic to WT iBMDMs +/- KLA, and quantified TNF after 6.5 hrs. We did not find any increases in TNF from any antibiotic alone or in combination with KLA (Fig. S6A).”

6. Figure 2D: The x axis labels are not aligned. Also, it would be nice to show this same figure except using dose-dependent killing with ciprofloxacin as a way to show that TNF concentration is proportional to amount of bacterial killing, regardless of mechanism. If this is not the case, it would reveal that other factors may be in play.

We have better aligned the x axis labels in Fig 2D. Doing the same experiment using ciprofloxacin to generate a gradient of different proportional amounts of viable bacteria would, in theory, be a good complement to Fig 2D. However, *in vitro* ciprofloxacin treatments at different doses do not generate the same gradations of viable bacteria as do different tetracycline doses. Instead, we find a minimum concentration above which nearly all to all bacteria are killed, and below which roughly equivalent amounts of bacteria survive. We provide that data below in Reviewer Response Figure 1.

Reviewer Response Figure 1: Surviving bacteria after treatment with the indicated concentration of ciprofloxacin for 6.5 hours.

7. Line 252 – 262: If more DNA is liberated into the soluble fraction with most of the cidal drugs, I think there should be some discussion on why we don't see an increase in TNF levels in macrophages treated with the cidal supernatant. One would think that there should be more "inflammatory soluble factors" in the form of the liberated DNA with the LPS levels being roughly equivalent.

This is an intriguing question. We hypothesize that there are two possible reasons for this discrepancy: 1. LPS is more completely liberated into solution from the antibiotic treated bacteria than is DNA. Although the cidal treatments are resulting in released bacterial DNA (Fig 3C), we believe (and our data in Fig 5A support) that much of it is still stuck to the dead/dying bacilli, so likely less able to pass through the filter than released LPS. 2. It is not entirely clear that immortalized macrophages are able to respond to bacterially derived extracellular DNA (eDNA) alone at the membrane – because the corresponding TLR9 sensor is localized to the endosome. Thus, even if more bacterially derived eDNA is coming through the filter with the cidal treatments, when the larger intact/dying bacilli are excluded by the filter, the macrophages may not generate a large response because the DNA on its own would not be phagocytosed and presented to TLR9 in the same manner as dying bacteria and eDNA together.

We have updated the indicated line to read "*independently inflammatory soluble factors*" (line 286).

We have also added the following to our description of Fig 5A in results: "*These images suggest that much of the cidal drug treated, released DNA remained tightly associated with the dead/dying bacilli (Fig 5A), possibly accounting for why we did not observe increased TNF responses from macrophages infected with filtered, cidal treated bacteria (Fig 3E).*" (lines 369-372).

8. Line 336: Why do we see increased extracellular DNA for cipro in this figure compared to the supernatant in the previous figure 4C? Is it just a difference in the timepoint?

We believe the reviewer is correct, and that this is likely a difference in the timepoint assessed: bacteria treated for 3 hrs on the gel vs. bacteria treated for 6 hrs in the imaging assays. We attempted treat for a longer period with the gel assay but ran into issues with DNA sample purity from the extracts (for the gel) that we did not have with the more self-contained imaging assays, where we stained in the infection environment.

9. Line 417: antibiotics is misspelled.

We have corrected this error, (see changed text on line 467).

10. Line 441 - 447: Are there any clinical meta-analyses comparing human outcomes in infection-mediated systemic inflammatory response syndrome or the like addressed with a static broad-spectrum antibiotic(s) over a cidal one?

To our knowledge the major one in the field is the Wald-Dickler meta-analysis we cite (Ref 28). It finds overwhelmingly (across 49 randomized controlled trials (RCTs) that – when prescribed appropriately, ie, to patients whose infections are susceptible to the relevant drug by MIC testing – both cidal and static drugs are equally effective for treating skin and soft tissue infections, bacterial pneumonia, typhoid fever/salmonellosis, and a few "other infections". In six RCTs there was a slight benefit to static treatments, and in just one RCT there was a slight benefit for the cidal treatment.

It is more challenging to assess this association in more intensive systemic infections, because patients with the highest acuity infections have nearly always received multiple different antibiotics.

11. It is important to recognize in this experiment that the mice were being infected suddenly with a large bacterial inoculum and just as suddenly treated with a large antibiotic dose. I do wonder how that might differ from mice that get sick more gradually that have been enduring a growing infection for hours or days. This could be worth mentioning or otherwise addressing in the discussion.

This is an important point. We have amended the discussion to directly note it (see lines 459-464 and below). Although chronic infections and extended establishment timeline infections were beyond the scope of the present paper, we would also

appreciate further work to identify how treating them with different kinds of antibiotics does/does not alter host immune responses.

New discussion text reads: “The magnitude of this phenotype emphasizes the potential therapeutic importance of understanding how a given antibiotic impacts downstream host immune responses to an acute, quickly treated infection (in addition to its impact on bacterial clearance). How different types of treatments impact chronic bacterial infections, or infections that take longer to identify and where treatment is initiated later during the course of infection, is a worthwhile area of future investigation.”

Comments from Reviewer 3:

Here, Gross et al undertake a careful and elegant study unpicking the role of released bacterial DNA in damaging inflammation following bactericidal drug treatment. Using an in vivo mouse model of acute peritonitis, the authors noted a disparity in the efficacy of treatment with cidal versus static drugs – with static drugs considerably more protective than cidal. Further studies confirmed that bacteria treated with cidal drugs induce higher levels of cytokines, particularly TNF (as a marker for inflammation), from infected macrophages. This inflammatory response is associated with bacterial DNA released following cidal treatment and is dependent upon the TLR-9 signalling pathway. This TLR-9 mediated inflammatory response is correlated with the failure of cidal drugs to treat the acute model of infection supported by the fact that efficacy of these drugs improves dramatically in TLR-9 null mice. These studies are generally well carried out and have broad implications for the future selection of drug treatments for high level bacterial infections. The authors themselves note that their observations are from a single in vivo model of infection – so they may want to be slightly more careful and tone down the extrapolation of this hypothesis more broadly. In addition, I have some further point that they should address.

We thank the reviewer for their supportive comments.

1. Can the authors please expand on the specifics/mechanism of how they think elevated TNF results in cidal drug failure?

Thank you for this question. To clarify: we are not claiming that the cidal drug treatment failed to kill the bacteria in our infection model; we see considerable – and equivalent – reductions in bacterial CFUs after both antibiotic treatments (see new Fig 1C/6C, reproduced on page 4). In this case, we are referring to treatment failure considering the survival outcomes of the different therapies for the mice. In other words: despite the similar efficacy of both drug treatments in reducing bacterial burden, the cidal treated mice still died at much higher rates than the static treated mice (Fig 1B). We believe this failure of the cidal drug treatment to protect mice at the same level as static drug treatment results from a hyper-inflammatory response to the cidal treated bacteria.

We use TNF throughout the paper as a representative biomarker of macrophage activation and systemic inflammation. In addition to the TNF elevation we consistently observe with cidal treatments, we also see the levels of several other

pro-inflammatory cytokines increasing more with these cidal treatments (see Fig S4-S5 for *in vitro* data, and Fig 1F for *in vivo* data). Taken together, we believe it is the increased overall systemic inflammatory responses that these cidal drug treatment induced cytokines potentiate that leads to the detrimental pathogenesis outcomes we observe in cidal drug treated WT mice. Mechanistically, it appears that this happens because only cidal drugs that kill the bacteria cause dead/dying bacteria to release DNA that macrophages sense and respond to via TLR9; static drugs do not have this effect (see Figs 4, 5 for *in vitro* data and Fig 6 for *in vivo* data).

2. As a non-immunologist – can you explain if TLR9 is known to “sense” anything outside of DNA? This may help non-specialist readers.

TLR9 is generally thought to sense only DNA and DNA containing immune complexes in the endosome. To our knowledge, there is no existing evidence that it senses other bacterial ligands. We have added references 36 and 37 as additional evidence of the DNA-sensing function of TLR9, and more background for interested, non-immunologist readers about ligands for all TLR receptors.

3. Treatment with static drugs to prevent a damaging immune response following destruction of a pathogen is not a novel concept. For instance, drugs that target eggs in schistosomiasis infections are contraindicated due to the damaging immune response that may result from a release of antigens. The authors may want to include some examples of such phenomenon in their discussion.

We thank the reviewer for bringing parallels from the world of parasitology to our attention. The one that strikes us as most germane to our investigation of how anti-infective drugs may impact host innate effector responses to treated infections is the case of loa loa, a filarial nematode parasite that causes loiasis in humans.

There are three primary drugs used to treat this parasite: diethylcarbamazine (DEC), ivermectin, and albendazole. DEC acts by altering arachidonic acid metabolism in both the parasite and in host vasculature in ways that ultimately promote parasite clearance by host eosinophils. By contrast, ivermectin and albendazole paralyze the parasites directly (by targeting parasite-specific ion channels and beta-tubulin respectively). Although all three are cidal in the sense that they result in parasite death in combination with host immune mechanisms, DEC has the strongest association in the literature with specifically driving host innate immune (eosinophil) activation.

In this context, it is instructive to note that DEC administration is the standard of care for definitive treatment of loiasis – but with an important caveat: it is contraindicated when patients present with a high parasitic load. DEC is contraindicated for these patients because of a documented association with potentially fatal encephalopathy complications specifically in patients with high parasite loads. Instead, for these patients, physicians first must reduce parasite burden down to appropriate levels by apheresis (physical removal), or treatment with ivermectin or albendazole before it is considered safe to begin DEC treatment.

This treatment protocol is an additional example of the clinical importance of accounting for innate immune responses to drug treatments and tailoring those treatments to account for known exacerbations of innate responses. We have updated the discussion to include this parallel in the context of independently immunomodulatory anti-infective drugs (see lines 466-467 and 477-487, new references (8-12 here; 63-67 in the paper), and below).

New discussion paragraph reads: “Another consideration is that some anti-infective drugs may themselves influence innate immune responses... An additional example of an anti-infective drug that is known to directly influence host innate immune reactions in important ways is diethylcarbamazine (DEC) therapy for loiasis. Unlike tetracycline that can dampen host responses, DEC is known to activate host eosinophils to facilitate parasite clearance,⁸ and it is the standard of care for definitive treatment of loa loa parasite infection in humans.⁹ However, it is contraindicated for patients with high parasite loads because of known associations with potentially fatal encephalopathy complications.¹⁰⁻¹² Instead, physicians treat patients presenting with high parasite loads with therapies that lower parasite burdens without activating host eosinophils (such as apheresis or other anti-parasitic agents that directly target parasite proteins to paralyze them for clearance),⁹ highlighting an additional high pathogen load infection context in which host innate responses play a critical role in determining the success or failure of the treatment.”

Though ivermectin may indeed function as a “static” anti-parasitic in paralyzing, but not directly killing loa loa, the mechanism of subsequent parasite death is not fully clear. What may be important is the slower kinetics of killing. We were not able to identify any other examples of static acting anti-parasitic agents but are open to adding additional related examples if the reviewer can help us to identify them.

4. Why did the authors switch between cEC1 and K12 in the experiments shown in Figs 1 and 2?

We conducted the initial screening experiment in Figure 2 before the *in vivo* work in Figure 1. As the project developed, we transitioned away from K12 (a model, laboratory *E. coli* strain) to cEC1 (a patient derived strain). If the reviewer feels the initial screen data detracts from the paper as a result of switching the strains, we will move it to the supplement or remove it.

5. Fig 4 – did the authors attempt to quantify the bacterial DNA released? Would be interesting to know a ballpark/bacteria. One could argue that such a number would be useful in deciding when and when not to apply cidal/static drugs for treatment.

We did attempt to quantitate the DNA in absolute ng/mL terms (in addition to the relative quantitation included in Fig 4C-D), using both nanodrop and tape station machines. Unfortunately, neither approach was successful: DNA from antibiotic treated bacteria – even when crudely precipitated with sodium acetate and ethanol – still contained high levels of impurities that led to unreliable measurements using these instruments designed for very pure nucleic acid samples.

Minor

1. In Table 1 (or S1) – called both in the text – can the authors add which drugs are cidal versus static and also their precise mechanism of action (ie inhibits protein synthesis).

We have corrected this to consistently refer to the table as Table 1, and added the requested columns to list the category of each drug and describe the mechanisms of action.

Response to Reviewers Reference List:

1. Ocampo, P. S. *et al.* Antagonism between Bacteriostatic and Bactericidal Antibiotics Is Prevalent. *Antimicrob. Agents Chemother.* **58**, 4573–4582 (2014).
2. LEPPER, M. H. & DOWLING, H. F. TREATMENT OF PNEUMOCOCCIC MENINGITIS WITH PENICILLIN COMPARED WITH PENICILLIN PLUS AUREOMYCIN: Studies Including Observations on an Apparent Antagonism Between Penicillin and Aureomycin. *AMA Arch. Intern. Med.* **88**, 489–494 (1951).
3. Blander, J. M. & Medzhitov, R. Toll-dependent selection of microbial antigens for presentation by dendritic cells. *Nature* **440**, 808–812 (2006).
4. Natarajan, M., Lin, K.-M., Hsueh, R. C., Sternweis, P. C. & Ranganathan, R. A global analysis of cross-talk in a mammalian cellular signalling network. *Nat. Cell Biol.* **8**, 571–580 (2006).
5. Lin, B., Dutta, B. & Fraser, I. D. C. Systematic Investigation of Multi-TLR Sensing Identifies Regulators of Sustained Gene Activation in Macrophages. *Cell Syst.* **5**, 25–37.e3 (2017).
6. Fisch, D. *et al.* Molecular definition of the endogenous Toll-like receptor signalling pathways. *Nature* **631**, 635–644 (2024).
7. Shiokawa, D. & Tanuma, S. Characterization of Human DNase I Family Endonucleases and Activation of DNase γ during Apoptosis. *Biochemistry* **40**, 143–152 (2001).
8. Maizels, R. M. & Denham, D. A. Diethylcarbamazine (DEC): immunopharmacological interactions of an anti-filarial drug. *Parasitology* **105**, S49–S60 (1992).
9. CDC. Clinical Treatment of Loiasis. *Filarial Worms* <https://www.cdc.gov/filarial-worms/hcp/clinical-care/loiasis.html> (2024).
10. Chippaux, J. P., Boussinesq, M., Gardon, J., Gardon-Wendel, N. & Ernould, J. C. Severe adverse reaction risks during mass treatment with ivermectin in loiasis-endemic areas. *Parasitol. Today Pers. Ed* **12**, 448–450 (1996).
11. Dieki, R., Nsi-Emvo, E. & Akue, J. P. The Human Filaria Loa loa: Update on Diagnostics and Immune Response. *Res. Rep. Trop. Med.* **13**, 41–54 (2022).
12. Carne, B., Boulesteix, J., Boutes, H. & Puruehnce, M. F. Five Cases of Encephalitis during Treatment of Loiasis with Diethylcarbamazine. (1991) doi:10.4269/ajtmh.1991.44.684.